# Reviews and Syntheses: Best practices for the application of marine GDGTs as proxy for paleotemperatures: sampling, processing, analyses, interpretation, and archiving protocols

- Peter K. Bijl¹\*, Kasia K. Śliwińska²\*, Bella Duncan³, Arnaud Huguet⁴, Sebastian Naeher⁵,⁶, Ronnakrit Rattanasriampaipong⁻,⁶, Claudia Sosa-Montes de Oca⁵, Alexandra Auderset¹⁰, Melissa A. Berke¹¹, Bum Soo Kim¹², Nina Davtian¹³, Tom Dunkley Jones¹⁴, Desmond D. Eefting¹⁵, Felix J. Elling¹⁶, Pierrick Fenies¹⁷, Gordon N. Inglis¹⁰, Lauren O'Connor¹, Richard D. Pancost⁵, Francien Peterse¹, Addison Rice¹, Appy Sluijs¹, Devika Varma¹⁶, Wenjie Xiao¹⁶, Yi Ge Zhang²⁰
  - <sup>1</sup>Department of Earth Sciences, Faculty of Geosciences, Utrecht University, Utrecht, the Netherlands. ORCID 000-000-002-1710-4012 (Bijl) 0000-0003-1872-1853 (O'Connor) 0000-0001-8781-2826 (Peterse) 0000-0002-4507-6717 (Rice) 0000-0003-2382-0215 (Sluijs)
  - <sup>2</sup>Geological Survey of Denmark and Greenland (GEUS), Department of geoenergy and storage, Copenhagen, Denmark. ORCID 0000-0001-5488-8832
- ORCID 0000-0001-5488-8832

  3Antarctic Research Centre, Victoria University of Wellington, Wellington, New Zealand. ORCID 0000-0003-1108-6033

  4Sorbonne Université, CNRS, EPHE, PSL, UMR METIS, Paris, 75005, France. ORCID 0000-0002-6124-2922

  Lincoln University, Department of Soil and Physical Sciences, PO Box 85084, Lincoln 7647, New Zealand ORCID 0000-0002-5336-6458
- School of Geography, Environment and Earth Sciences, Victoria University of Wellington, Wellington, New Zealand,
   University Corporation for Atmospheric Research, Boulder, CO, 80307 USA. ORCID 0000-0002-1425-8737
   Department of Geosciences, The University of Arizona, Tucson, AZ, 85721 USA
   Organic Geochemistry Unit, School of Earth Sciences, School of Chemistry, The Cabot Institute for the Environment,
  - Organic Geochemistry Unit, School of Earth Sciences, School of Chemistry, The Cabot Institute for the Environment, University of Bristol, Bristol, UK. ORCID 0000-0003-4451-6458 (Sosa-Montes de Oca) ORCID: 0000-0003-0298-4026 (Pancost)
  - <sup>10</sup>School of Ocean and Earth Science, University of Southampton, Southampton SO14 3ZH, United Kingdom. ORCID: 0000-0002-6316-4980
  - <sup>11</sup>Department of Civil and Environmental Engineering and Earth Sciences, University of Notre Dame, Notre Dame IN 46556, USA. ORCID: 0000-0003-4810-3773
- 30 <sup>12</sup>Amentum, JSC Engineering and Technical Support (JETS) Contract, NASA Johnson Space Center, Houston, TX 77058, USA. ORCID: 0000-0002-0533-1330
  - <sup>13</sup>CEREGE, Aix-Marseille Université, CNRS, IRD, INRAE, Collège de France, Technopôle de l'Arbois, 13545 Aix-en-Provence, France. ORCID: 0000-0002-3047-6064
  - <sup>14</sup>School of Geography, Earth and Environmental Sciences, University of Birmingham, B15 2TT, United Kingdom. ORCID 0000-0002-9518-8143
    - <sup>15</sup>GeoLab, Faculty of Geoscience, Utrecht University, Utrecht, the Netherlands





- <sup>16</sup>Leibniz-Laboratory for Radiometric Dating and Isotope Research, Christian-Albrechts-University of Kiel, 24118 Kiel, Germany, ORCID: 0000-0003-0405-4033
- <sup>17</sup>Institute of Oceanography, National Taiwan University, No. 1, Sec. 4, Roosevelt Road, 10617 Taipei, Taiwan. ORCID: 0000-0002-1183-9704
- <sup>18</sup>Department of Marine Microbiology and Biogeochemistry, NIOZ Royal Netherlands Institute for Sea Research, Den Burg, the Netherlands. ORCID 0000-0001-9707-6690 (Varma)

<sup>19</sup>Department of Biology, HADAL & Nordcee, University of Southern Denmark, 5230 Odense M, Denmark. ORCID: 0000-0002-4734-683X

<sup>20</sup>Guangzhou institute of Geochemistry, Chinese Academy of Sciences, Guangzhou, China. 0000-0001-7331-1246

\* Both contributed equally to this work

Correspondence to: Peter K. Bijl (p.k.bijl@uu.nl) and Kasia K. Sliwinska (kksl@geus.dk)

Abstract. Marine glycerol dialkyl glycerol tetraethers (GDGTs) are used in various proxies (such as TEX<sub>86</sub>) to reconstruct past ocean temperatures. Over 20 years of improvements in GDGT sample processing, analytical techniques, data interpretation and our understanding of proxy functioning have led to the collective development of a set of best practices in all these areas. Further, the importance of Open Science in research has increased the emphasis on the systematic documentation of data generation, reporting and archiving processes for optimal reusability of data. In this paper, we provide protocols and best practices for obtaining, interpreting and presenting GDGT data (with a focus on marine GDGTs), from sampling to data archiving. The purpose of this paper is to optimize inter-laboratory comparability of GDGT data, and to ensure published data follows modern open access principles.

#### Acronyms:

APCI: atmospheric pressure chemical ionization

AOM: anaerobic methane oxidation

ANME: anaerobic methanotrophic

ASE: accelerated solvent extraction

BIT: branched and isoprenoid tetraether

brGDGT: branched glycerol dialkyl glycerol tetraether

CCSF: core composite depth below sea floor

CL: core lipid


CN: cyano column

cren: crenarchaeol

cren': crenarchaeol stereoisomer

CSF: core depth below sea floor

DCM: dichloromethane

GP: gaussian process

GDGT: glycerol dialkyl glycerol tetraether

GDD: glycerol dialkyl diether

GMGT: glycerol monoalkyl glycerol tetraether

GTGT: glycerol trialkyl glycerol tetraether

HPLC: high-performance liquid chromatography

HPLC-APCI-MS: high performance liquid chromatography-atmospheric pressure chemical ionisation-mass spectrometry

IODP: International Ocean Discovery Program

IPL: intact polar lipid

isoGDGT: isoprenoid glycerol dialkyl glycerol tetraether

LC-MS: liquid chromatography-mass spectrometry

MI: methane index MeOH: methanol

OH-GDGT: hydroxylated glycerol dialkyl glycerol tetraether

OM<sub>soil</sub>: soil organic matter

PFTE: polytetrafluoroethylene (commonly known as Teflon)

RI: ring index






RSE: relative standard error

SIM: selected ion monitoring

SPM: suspended particulate matter

SST: sea surface temperature subT: subsurface temperature

TEX<sub>86</sub>: tetraether index of 86 carbon atoms

95 TLE: total lipid extract TOC: total organic carbon

#### 1 Introduction

Glycerol Dialkyl Glycerol Tetraethers (GDGTs) are membrane lipids that are widely applied as indicators of past water and air temperature, soil organic matter (OM<sub>soil</sub>) input in marine settings, as well as soil pH (Schouten et al., 2013a; Fig. 1). They are synthesized by a group of archaea (the main source in the ocean is assumed to be from *Nitrososphaera*; formerly Thaumarchaeota and Crenarchaeota; De Rosa and Gambacorta, 1988; Koga et al., 1993) and bacterial groups (Weijers et al., 2006b), including Acidobacteria and likely additional sources (Weijers et al., 2009, 2010; Sinninghe Damsté et al., 2011; Halamka et al., 2021, 2023; Chen et al., 2022). The GDGT 'pool' includes isoprenoid (isoGDGTs), and branched GDGTs (brGDGTs). Although iso- and brGDGTs are synthesized in both terrestrial and marine settings, isoGDGTs are typically associated with marine production (Schouten et al., 2000) and brGDGTs with terrestrial sedimentary settings (Sinninghe Damsté et al., 2000). The isoGDGTs are characterized by their isoprenoid carbon skeleton and includes isoGDGTs-0 to -8 (where numerals refer to the number of cyclopentane moieties), and crenarchaeol (and a stereoisomer of that), which has four cyclopentane moieties and one cyclohexane ring (De Rosa and Gambacorta, 1988; Sinninghe Damsté et al., 2002b). It has been shown that the level of cyclization in marine isoGDGTs is correlated to mean annual sea surface or subsurface temperature (Schouten et al., 2002), and represents a powerful paleotemperature proxy.

In addition to iso- and brGDGTs, other variants of GDGTs have also been identified, including hydroxylated GDGTs (OH-GDGTs) (Liu et al., 2012c) as well as the much less studied glycerol trialkyl glycerol tetraethers (GTGTs; e.g., De Rosa and Gambacorta, 1988), glycerol dialkyl diethers (GDDs; which are not membrane-spanning lipids (Coffinet et al., 2015; Mitrović et al., 2023; Hingley et al., 2024)), and glycerol monoalkyl glycerol tetraethers (GMGTs) (Morii et al., 1998; Liu et al. 2012a; Naafs et al., 2018; Baxter et al., 2019).

BrGDGTs have an alkyl backbone to which a total of four to six methyl branches can be attached, and can contain up to two cyclopentane moieties (Sinninghe Damsté et al., 2000; Weijers et al., 2006b). They are typically produced in terrestrial settings and therefore much less common in marine settings than the isoGDGTs. The BrGDGTs from soils are used to reconstruct continental air temperatures and soil pH (Weijers et al., 2007b; De Jonge et al., 2024), including, in some cases, after transport to and deposition in marine depositional settings (e.g., Weijers et al., 2007a; Pross et al., 2012; Pancost et al., 2013; Śliwińska et al., 2014; De Jonge et al., 2014b; Bijl et al., 2018, 2021; Willard, 2019; Dearing Crampton-Flood et al., 2018). The abundance of brGDGTs relative to that of crenarchaeol in marine sediments is further used to reconstruct input of soil material to the marine environment as part of the Branched and Isoprenoid Tetraether (BIT) index (Hopmans et al., 2004), albeit complicated by the in situ production of marine brGDGTs (Peterse et al., 2009; Sinninghe Damsté, 2016), as well as the production of crenarchaeol in soils (Weijers et al., 2006a). A series of studies over the past years have demonstrated that factors other than temperature affect the distribution of GDGTs in sediments (see e.g., Schouten et al., 2013a for a review, and Bijl et al., 2021 for application). While these other factors could potentially affect the reliability of GDGTs as a temperature proxy, these effects can now be detected, and they yield additional information on oceanographic and environmental conditions.

The correlation between isoGDGTs and SST in marine sediments was first described by Schouten et al. (2002), who proposed the TetraEther indeX consisting of 86 carbon atoms, known as TEX<sub>86</sub>, to quantify the degree of cyclization of isoGDGTs. The shape of the relationship between TEX<sub>86</sub> and SST has been explored using a growing dataset of isoGDGTs in surface sediment spanning the modern ocean (Fig. 1), and temperatures of the overlying sea surface and subsurface waters (Schouten et al., 2002; Liu et al. 2009; Kim et al., 2010; Tierney 2nd Tingley, 2015). Together with other geochemical proxies, both organic-(alkenones, diols) and calcite- ( $\delta^{18}$ O,  $\Delta_{47}$  and Mg/Ca measured on planktic foraminifera) based, TEX<sub>86</sub> is a widely applied paleotemperature proxy. Paleo-applications of the proxy focus on the analysis of core lipids (CLs), i.e., GDGTs without their polar head groups, as those are rapidly lost during burial diagenesis. Recently, OH-GDGTs have shown to have higher sensitivity in a lower temperature range than isoGDGTs (Huguet et al., 2013; Lü et al., 2015; Varma et al., 2024a, 2024b). The relative abundance of OH-GDGTs over isoGDGTs, as well as the relative abundance of the different OH-GDGTs show promising relationships to temperature (Huguet et al., 2013; Fietz et al. 2016; Varma et al. 2024b). Specifically, the addition of OH-GDGT-0 to the denominator of the TEX<sub>86</sub> leads to an improved temperature sensitivity at the cold end (.e., 

Figure 1: Global map of surface sediment (core top) samples for isoprenoid GDGTs is now composed of data retrieved from previous major compilation efforts (i) published during 2002-2014 (light grey) (Kim et al., 2008, 2010; Tierney and Tingley, 2014, 2015 and references therein) and (ii) published since 2015 (dark grey) (Kaiser et al., 2015; Kim et al., 2015; Rodrigo-Gámiz et al., 2015; Tierney et al., 2015b; Kusch et al., 2016; Pan et al., 2016; Richey and Tierney, 2016; Sinninghe Damsté, 2016; Jaeschke et al., 2017; Ceccopieri et al., 2018; Chen et al., 2018; Lo et al., 2018; Schukies, 2018; Yang et al., 2018; Lamping et al., 2021; Harning et al., 2019, 2023; Wei et al., 2020 Sinninghe Damsté et al., 2022; Hagemann et al., 2023; Varma et al., 2024b; Rattanasriampaipong et al., 2025)—along with the location of all TEX86 records in PhanSST (Judd et al., 2022 and references therein) in the colour of the dominant geologic time interval represented.

Developments in Open Science in academia have increased awareness of scientific integrity in data reporting. The best-practice of reporting scientific data follows FAIR (Findable, Accessible, Interoperable, and Reusable; Wilkinson et al., 2016) open science principles, which were later translated into specific open access guidelines and objectives for the geosciences community into ICON (Integrated, Coordinated, Open and Networked; Goldman et al., 2022). The advantages of applying these principles are profound: they systematize the presentation of generated data that is connected to publications and ensure that the data (including metadata) are correctly reported. In the longer term, the availability of properly archived data facilitates and stimulates its reuse by stakeholders, both in- and outside academia, with the generation of new insights far into the future. A clear example of such an effort is the SST data compilation PhanSST (Judd et al., 2022; Fig. 1), which supports the community by consolidating paleotemperature proxy data in a single, easyly searchable repository. The PhanSST dataset is op open-access free-to-use portal, presented in a standarised format to ensure interoperabilibity. It os accompanied by available metadata on samples, sites and sources. The FAIR Open Science principles add value to the generated data and have ramifications for how the community best report these. We however conclude that for isoGDGT analyses, the community has yet to agree on the common framework for data presentation, and this paper is a step towards that ambition.

The most common workflow in GDGT analysis consists of (1) sample selection and collection and (2) storage, (3) extracting the lipids in a total lipid extract (TLE) (4) TLE clean up procedures using column chromatography, (5) GDGT analysis using (Ultra) High Performance Liquid Chromatography—Mass Spectrometry ((U)HPLC—MS), (7) GDGT peak integration, (8) data interpretation, (9) data reporting and archiving. In this paper, we review and summarize best practices to generate, report and archive marine GDGT data, following the steps above. We base our review on examples from the literature, empirical studies, and provide data that underpins over 20 years of experience in the biomarker community. This initiative was put forward during a GDGT workshop in Zurich in 2023, where a large part (albeit with a notable bias towards European participants) of the GDGT community gathered to discuss the steps forward in proxy understanding, development and application. The purpose of this paper is to optimize inter-laboratory comparability of marine GDGT data that is produced by laboratories worldwide, and to ensure published data follows modern FAIR open access principles.

## 2 Sampling






## 2.1 Types of samples

Marine sediment samples for GDGT analysis can be obtained from sediment cores - mainly drill cores, piston/gravity cores, multicore samples, grab samples - or outcrops. For studies on modern GDGTs, water column sampling is commonly undertaken using sediment traps or with Niskin bottles and laboratory filtration, or using *in situ* pumps to collect Suspended Particulate Matter (SPM) from the water column. As this paper focuses on sedimentary GDGTs, we do not discuss these further.

Surface and subsurface marine sediment sampling (coring) happens via local/national seagoing cruises and expeditions, whereas the International Ocean Discovery Program (IODP) and its legacy programs Deep Sea Drilling Project (DSDP) and Ocean Drilling Program (ODP), are the only research programs to have recovered fully cored marine stratigraphic records (up to 2 km of sediment) across a full range of ocean depths (up to 8 km water depth). The IODP uses dedicated research vessels and extensive international collaborations (e.g., the IODP Science plan: <a href="https://www.iodp.org/science-plan/127-low-resolution-pdf-version/file">https://www.iodp.org/science-plan/127-low-resolution-pdf-version/file</a> and the 2050 IODP science framework: <a href="https://www.iodp.org/docs/iodp-future/1086-2050-science-framework-full-document/file">https://www.iodp.org/docs/iodp-future/1086-2050-science-framework-full-document/file</a>). Depending on the location of the drilling, IODP cores are stored and curated at specific repositories at the University of Bremen (Germany), Gulf Coast Repository at Texas A&M University (USA) and Kochi Core Center at Kochi University/JAMSTEC (Japan), see <a href="https://www.iodp.org/resources/core-repositories">www.iodp.org/resources/core-repositories</a>). In all three repositories sediment cores are stored in a reefer (refrigerated storage area) maintained at a temperature of +4 °C and controlled humidity (<a href="https://www.iodp.org/resources/core-repositories">www.iodp.org/resources/core-repositories</a>). These repositories ensure availability of high-quality core material to member states long after the completion of each expedition. Many national research facilities have repositories of cores and marine samples as well, but the storing capacity and conditions can vary. Moreover, most sediment core repositories are not equipped to curate processed subsets of core samples, which leaves the responsibility for curating lipid biomarker fractions and TLEs to the respective geochemistry labo

In order to study marine successions on land, two options are possible: onshore drilling for obtaining a continuous core record or hand-sampling from outcrops. Sampling outcrops for marine GDGTs may lead to challenges related to the preservation of the compounds. Sedimentary deposits on land, as in well-ventilated ocean basins, are typically exposed to various degrees of oxic degradation (Huguet et al., 2008; Lenger et al., 2013). Furthermore, outcrop exposure can alter the autochthonous GDGT signal through inputs from modern GDGTs (from e.g., modern soils) to an unknown amount, but this has never been quantified. For outcrop sampling it is thus crucial to retrieve fresh sediment material that is unaffected by weathering. The depth of the

unweathered sediment in outcrops varies spatially and can reach up to tens of meters deep (Clyde et al., 2013) often depending on the characteristics of the sediment and local climatic conditions (Strakhov, 1967).

Moreover, deposits can be affected by variable degrees of heating and pressure which, if too intense, could affect GDGT preservation. GDGTs have been shown to thermally degrade at temperatures >260 °C (Schouten et al., 2004; although only one sample was studied – this threshold might be dependent on sample type), making the thermal history of especially older sediments an important consideration in the interpretation of their GDGT content. The reaction kinetics of GDGTs are mainly governed by temperature and the duration of heating, making older sediments more susceptible to the effects of thermal alteration. An understanding of the thermal history, including contact metamorphism from igneous intrusions or lava flows, is possible through an estimation of the thermal maturity of organic matter in sediment samples by using either optical (e.g., Thermal Alteration Index, vitrinite reflectance) or geochemical analysis of organic matter (e.g., RockEval Pyrolysis), or an assessment of the hopane composition, and their stereochemical configuration (e.g., Schouten et al., 2004).

In thermally mature sediments, the concentration of the original/autochthonous GDGT pool may either i) become biased due to preferential preservation of GDGTs, with TEX<sub>86</sub> decreasing with increasing maturity (although this is from just one study and awaits further confirmation by others; Schouten et al., 2004, 2013a), where brGDGTs are preserved preferentially over isoGDGTs thus affecting the BIT index (Huguet et al., 2008; Schouten et al., 2013a), or ii) may decrease to below detection limit, even when the total organic carbon (TOC) content remains high (Schouten et al., 2004). As temperature and pressure increases with burial depth, there is a risk that GDGTs degrade at too deep depths. To date, the deepest-buried GDGTs were retrieved from North Atlantic Deep Sea Drilling Project Sites 398 and 603, from immature Lower Cretaceous sediments at depths of over 1.5 km below sea floor (Littler et al., 2011; Naafs and Pancost, 2016). This provides evidence that GDGTs can be preserved even in relatively deeply buried strata, as long as sediments do not surpass the temperature threshold of 260 °C.

#### 2.2 Contamination, storage, and initial processing of environmental samples







Best-practice sampling seeks to avoid any addition of organic biomarkers or compounds to the target sample from external sources (e.g., Brocks et al., 1999). Although this paper only concerns GDGTs, sample integrity should be maintained for all lipid biomarkers where possible, also to maximize reuse of the extractions for other analyses. When coring sediments using drilling mud/fluid, organic geochemists, whether present on-site or on-board a drilling vessel, are recommended to sample and process drill fluid as a reference where possible. Many drilling muds might not contain biomarkers or GDGTs, but to minimize contamination risk, it is strongly recommended to avoid using the external part of the core that has been in contact with drilling mud/fluids. Even though oil-contamination may not have an effect on GDGTs (Kellner et al. accepted), it might affect other lipid biomarkers, in other fractions. For legacy cores, it is recommended to remove the outermost part of the core (which could

have been contaminated with recirculating drilling mud/fluid). Once the samples/cores are obtained, they need to be sealed from oxygen exposure and, to limit microbial activity, ideally stored in a cold room (4°C), or better yet, in a freezer (-20°C), or freeze-dried immediately. One example that underlines the importance of this comes from a recent study (Frieling et al., 2023), where the same lithological formation of Paleocene-Eocene (58–54 Ma) age was sampled at an outcrop and in ~50 year-old sediment cores, which were taken 3 km away from the outcrop site. The cores had been stored in a non-refrigerated, uninsulated storage facility in Australia. GDGT extractions carried out on the age-equivalent samples from both outcrop and cores revealed orders of magnitude differences in their GDGT yield (see data files connected to Frieling et al., 2023). It was concluded that the 5 decades of aerial exposure to temperature and humidity swings had degraded the organic matter (the dinoflagellate cysts (Frieling et al., 2018) as well as affected the GDGT concentrations (Frieling et al., 2023)) in the old cores, to the extent that dinocyst assemblages were significantly altered and GDGTs were no longer quantifiable. Albeit extreme, this example highlights the fact that proper storage is crucial for the longevity of the preservation of GDGTs in sample material.

Once collected from the core/outcrop, sediment samples are often stored in plastic bags, but this is not recommended for samples taken for biomarker analyses. Plastic contamination (Grosjean and Logan, 2007) can be even more problematic with older sample material, where plastic bags can disintegrate with time into microscopic flakes and mix with the sediment. Although contamination from plastic will not necessarily be an issue for analysing GDGTs since it is typically (though not always) conducted in selective ion monitoring (SIM) mode (i.e., coelution of contaminants will not be shown), it can cause analytical complications in other fractions (e.g., Smith et al., 1993), and thus it limits reuse of extracts for analyses of non-polar biomarkers. We recommend that new samples are either i) wrapped in aluminum foil and stored in a regular plastic sample bag, but note that aluminum foil also disintegrates over time or ii) stored in a plasticizer-free sample bag dedicated to storage of samples for organic geochemistry.

#### 3 Processing samples for GDGTs

#### 3.1 Drying samples







Prior to lipid extraction, any moisture or water content in the sediment sample is recommended to be removed for maximizing the extraction efficiency (i.e., it will allow solvents to better penetrate sediments during extraction) while preventing the loss of polar compounds bound to the water. Two sample drying methods are typically used when processing marine sediment samples: i) freeze drying - samples are dried under vacuum and below -60°C until dry; and ii) oven drying - samples are placed in a laboratory oven (or drying device) for overnight or longer. Even though GDGTs will not be affected by temperatures above 40°C, it is recommended to not exceed 40°C when drying samples in the oven, to prevent the degradation of sensitive

compounds that might occur at higher temperatures (Wiltshire and Du Preez, 1994; Rosengard et al., 2018). Freeze-drying is the preferred option (Rosengard et al., 2018). A study comparing impact of freeze-drying vs, air drying samples on the yield of long chain alkenones, showed a significant loss of alkenones for samples which were air-dried (McClymont et al., 2007), suggesting that GDGTs may likewise be re affected by oven-drying.

#### 3.2 Powdering samples






Aggregated or lithified sample materials, such as marine sediments, are usually homogenized (i.e., making sample materials into powder) to increase surface area, maximizing (lipid) extraction efficiency. Powdering of the sediment can be done either by hand in an agate mortar or ball mill (e.g., Austin, 1984; Richardson et al., 2002). The choice between these depends on the type and volume of the material: smaller volumes with low lipid yields might be better hand-ground carefully in a mortar, while large volume samples that are lithified, are best ground in a ball mill. In both methods it is crucial to clean the mortar/mill between every sample (using either: i) DCM/MeOH, ii) acetone, or iii) by grinding quartz sands that have been cleaned via combustion followed by a rinse with solvents) to minimize cross-over contamination. Experience shows that the extraction efficiency increases when samples are powdered to a finer size, and while this has—to the best of our knowledge—not been formally quantified for lipid biomarkers, it has been demonstrated in plant sicence (e.g., Alava et al., 2012).

#### 3.3 Extractions of lipids

The extraction of organic compounds (including lipids) from natural samples (sediment, biomass, etc) is a crucial first step in the analysis of biomarkers for paleoclimate research. The TLEs may contain thousands of various organic compounds, and several of them can be used as proxies for seawater temperature reconstructions, such as long-chain ( $C_{37}$ ) ketones (alkenones) ( $U_{37}^{Kr}$ ; Brassell et al., 1986), long chain diols (e.g., Rampen et al., 2012) and GDGTs (Schouten et al., 2002). Sample preparation steps for all these lipids-based proxies, including marine GDGT analysis, usually consists of lipid extraction followed by column separation. Different extraction techniques exist, and tests with archeological samples showed that the extraction method could affect the extraction yields and quality (e.g., Scherer et al., 2024). Below we describe and discuss the most common extraction techniques used for extraction of GDGTs, and compare their advantages and disadvantages. The amount of sediment to be extracted is generally inversely correlated to TOC content, i.e., smaller amounts are required from higher TOC samples. The sample size can therefore play a role in selecting the optimal extraction technique, as there is large overlap in solvent volume associated with each extraction method.

# 3.1 Accelerated Solvent Extraction (ASE)






The ASE techniqueis widely used to automatize lipid extraction from sediments, and compares well with Soxhlet extraction (e.g., Jansen et al., 2006). The typical sample size is between 8 and 50 grams (up to 100 ml), depending on the TOC content and sample type. Powdered sediments are typically mixed with either combusted quartz sand, diatomaceous earth, or glass wool (e.g., Huguet et al., 2010), to allow a better solvent flow through the cell and avoid clumping. Subsequently samples are packed into a metal extraction cell. Total lipids are extracted from the sample using a mixture of solvents, most commonly dichloromethane (DCM) and methanol (MeOH) in proportions 9 to 1 (v/v) under high temperature (commonly 100 °C) and high pressure (>7.6 × 106 Pa; Jansen et al., 2006; Huguet et al., 2006). Higher temperatures are used to enhance the efficiency and speed of the extraction process. Samples can be extracted multiple (i.e., 2–3) times to optimize the extraction yield (e.g., Lengger et al., 2012). The advantage of ASE is that it is a fully automated and fast extraction technique, but it uses relatively large volumes of solvent for extraction and requires proper cleaning procedures for the extraction cells (see best practices in cleaning in Section 3.7).

#### 3.3.2 Microwave-assisted extraction oven

The microwave-assisted extraction technique relies on the use of microwave energy applied to heat pressurized vessels containing solvent and powdered sediment. Samples, typically between ~5 and 7 g, are placed into glass vials held by Teflon tubes, before adding a mixture of DCM and MeOH (most commonly 9:1 or 3:1 (v/v); Schouten et al., 2013b), as for the ASE. The microwave is programmed for the extraction, with controlled heating of the solvent and sediment followed by cooling (e.g., Huguet et al., 2010 for examples of conditions). Like ASE, microwave extraction is fully automated, but it has an advantage using less solvent and reducing carry-over contamination as it does not require tubing between samples. The main disadvantage of microwave extraction relative to ASE is the additional step required to separate the solvent carrying total lipid extract (TLE) from the sediment. For smaller models of microwaves, the glass vials containing sediment and solvent can be placed in a centrifuge to separate the aliquot from the sediment. In the case of larger microwave models, the aliquot can be either transferred into a vial compatible with centrifugation or separated from the sediment by e.g., gently pipetting the aliquot or pouring the aliquot into a clean vial. The second option will however require some extra time (1 to several hours) to allow the sediment to settle.

ASE and microwave extraction methods yield similar results for GDGTs in terms of biomarker yield and extraction quality (see Supplementary Table S1). However, it should be noted that specific settings of each method may impact the outcomes. Comparisons between ASE and microwave extractions (Frieling et al., 2023) give comparable GDGT yield and quality with a microwave heat setting of  $70^{\circ}$ C, For the microwave-assisted extraction, significant biases in TEX<sub>86</sub> indices (higher by <0.2

units) can occur when the temperature is set at 100-110°C (see Supplementary Table S1). For both extraction methods, repeated extractions of the same sample increases extraction yield.

#### 3.3.3 Soxhlet





Soxhlet extraction is usually performed on samples with a low lipid yield compared to the previously described methods. The powdered sediment, sometimes with an admixture of quartz to prevent channeling, is placed into a pre-extracted cellulose thimble which is loaded into a Soxhlet apparatus; solvent is then heated until boiling (normally less than 70 °C for DCM:MeOH (2:1, v/v) mixture) under reflux for several hours/days (commonly 24 hrs; (e.g., Huguet et al., 2010; Naeher et al., 2014b) or more (up to 48 hrs) for old sediment or low TOC) and flows through the sample, allowing the extraction of lipids. As for the other techniques, the solvent mixture usually consists of DCM and MeOH (Schouten et al., 2013b). The advantage of this technique is that extraction can be performed ultra-clean, enabling a comprehensive recovery of lipids. The setup, however, is relatively slow (one instrument is used for extraction of between one to ten samples between 24 and 48 hours) takes considerable amount of space and requires large volumes of solvent for extraction and cleaning, and complete recovery of all extracted lipids is tedious, making it less efficient than ASE or microwave for large batches of samples.

#### 3.3.4 Ultrasonic extraction

Ultrasonic extraction method is usually applied for smaller samples (< 15 g) using an ultrasonic bath at ambient temperature, to avoid overheating solvents. Samples are commonly mixed with solvent (typically a DCM/MeOH mixture) for a short time (10–15 minutes, Yang et al., 2018) and subjected to centrifugation to retrieve the supernatant. The extraction is usually repeated (3 to 5 times) and the supernatants are combined, forming the total lipid extract containing GDGTs (e.g., Zhang et al., 2012; Yang et al., 2018). The final step includes a transfer of the TLE via a small Na<sub>2</sub>SO<sub>4</sub> column (pipette) into a preweighed vial.

## 360 3.3.5 Bligh and Dyer extraction

Bligh and Dyer extraction is a widely-used extraction protocol for isolating lipids from biological matrices. It is based on liquid-liquid phase extration with a ternary solvent system (i.e., chloroform, methanol, and water; Bligh and Dyer, 1959). Briefly, homogenized samples are initially added to a chloroform/methanol (e.g., generally 1:2 (v:v)) mixture, and this is vortexed to enhance the liquid extraction efficiency. The sample-solvent mixture is then filtered, or the supernatant is recovered upon centrifugation, to separate the solvent from the solid tissue or sediment. Subsequently, chloroform and water are introduced, vortexed, and this separates lipids from aqueous-dissolvable (i.e., sugars, proteins etc) phases based on partial miscibility between the two liquids. This extraction method is effective for extracting lipids from a variety of substrates. Yet,

it requires the use of toxic solvents, and potentially does not recover all lipids (also GDGTs;e.g., Huguet et al., 2010; Weber et al., 2017) from high-TOC samples.

# 3.3.6 Core versus intact polar lipid extraction

GDGT core lipids are traditionally extracted using one of the techniques mentioned above. The efficiency of these different extraction techniques has been compared in multiple studies (e.g., Huguet et al., 2010; Lengger et al., 2012; Schouten et al., 2013b). Schouten et al. (2007) compared Soxhlet, ultrasonic, and ASE extraction techniques for CL while Lengger et al. (2012) compared Soxhlet, Bligh and Dyer and ASE for extraction of both IPL and CL. Both studies showed that the extraction efficiency of these methods for CL were not significantly influenced by the applied method. In contrast, Huguet et al. (2010) suggested that CL GDGTs may be more efficiently extracted with ultrasonic or Soxhlet extraction than with ASE. Nevertheless, an extensive round robin study of TEX<sub>86</sub> and BIT analyses involving 35 laboratories (Schouten et al., 2013b) revealed that neither TEX<sub>86</sub> nor BIT index are substantially impacted by sediment workup (extraction and processing), indicating that any of the aforementioned techniques could be used for the determination of marine CL GDGT distributions. In contrast to CL, IPL extraction usually follows a gentler method of an ultrasonic extraction modified from Bligh and Dyer (1959). GDGT extraction efficiency in IPL was also suggested to be highly dependent on the applied extraction technique (Huguet et al., 2010). The modification of the Bligh and Dyer extraction protocol including the use of detergent was shown to increase the yield of archaeal lipids in cultures and marine suspended particulate matter compared to the original Bligh and Dyer methodology, even though no obvious change in extraction efficiency was observed for marine sediments (Evans et al., 2022). It commonly uses a solvent mixture of MeOH, DCM, and an aqueous buffer (2:1:0.8; v/v/v). Protocols for the extraction of IPL GDGTs based on the Bligh and Dyer method were compared (e.g., Huguet et al., 2010; Evans et al., 2022) and yielded accurate comparative qualitative data for different extraction methods. If the intact polar lipids need to be separated from core lipids, a modified protocol is needed (Pitcher et al., 2009; Lengger et al., 2012), where first the CL fraction is eluted with hexane/ethyl acetate 1:2 (v/v), followed by the IPL fraction with MeOH.

## 390 3.4 Drying of TLE





Regardless of the applied extraction method, the obtained TLEs need to be dried down by evaporating the excess solvent mixture to enable quantification. As exposure to oxygen at this step is to be avoided, solvent evaporation is typically carried away under flowing N<sub>2</sub>. For large volumes of solvent, the solvent can be removed using a distillation setup under vacuum (Rotavap), which lowers the boiling point of the solvents, whereas for smaller quantities the solvent is evaporated under flowing N<sub>2</sub> (Turbovap). In both cases it is recommended that the temperature does not exceed 25 °C, to prevent a loss of biomarkers or IPL headgroups. Notably, higher temperatures (>25°C) and long drying under N<sub>2</sub> may not affect GDGTs, but can cause the loss of (semi-)volatile compounds (e.g., pristane, phytane, short-chain alkanes and fatty acids).

## 3.5 Cleaning up - column separation techniques





In some cases, the TLE may contain traces of sediment residue (occasionally following microwave-assisted extraction). In this case it may be necessary to filter the TLE over a glass pipette with either i) extracted cotton or glass wool; ii) Na<sub>2</sub>SO<sub>4</sub> column, or a iii) extracted paper filter in a funnel. However, if the TLE is further separated into various fractions using column chromatography, the remaining sediment fraction will stay on the column and be separated from the lipid extract. In that case, removing the residue may not be strictly necessary, although the weight of the TLE will be overestimated due to the presence of the residue.

In extracts from anoxic marine sediments with high TOC content, elemental sulfur may need to be removed. To do this, acid-activated copper is added (Smith et al., 1984); either i) directly to the solvent (DCM:MeOH) during Soxhlet extraction, or ii) after extraction, to the TLE, and stirred overnight. However, this step can also be applied after the separation of the TLE into fractions (see below), and then to the specific fraction that contains the elemental sulfur.

In order to minimize the weardown of the LC column as well as to concentrate and analyze the abundance of GDGTs it is optimal to isolate these compounds using small column chromatography. The isoGDGTs, brGDGTs (Section 8.2) and OH-GDGTs (Section 8.1) (Fig. 3) generally elude in the polar fraction. The separation is typically achieved by passing the TLE over either activated alumina (e.g., Huguet et al., 2006), partly deactivated silica (e.g., Naeher et al., 2014a) or aminopropyl (e.g., Russell and Werne, 2007) columns as the stationary phase (see also Escala et al., 2009). Various solvent mixtures can be applied as mobile phases, and up to 6 fractions can be separated, depending on TLE content, and the desired level of fractination, isolation and purification. Commonly applied solvents are hexane or hexane/DCM (9:1 v/v) to obtain an apolar fraction and DCM/MeOH (1:1 v/v) to obtain a polar fraction. Depending on other compounds that may be of interest, additional fractions can be eluted. For instance, an intermediately polar fraction is eluted using hexane/DCM 1:1 or 1:2 (v/v) to obtain ketones (e.g., Grant et al., 2023), in particular long-chain ketones or alkenones which are alternative paleoclimate proxies for sea surface temperature estimation (i.e.,  $U_{37}^{K\prime}$  index; e.g., Prahl and Wakeham, 1987; Herbert, 2014). When the polar fraction shows many compounds co-eluting with the GDGTs, an additional fraction using an ethyl acetate/DCM solvent mixture (1:1 v/v) can be generated to further purify the polar fraction (e.g., Bijl et al., 2013). Deviating from these typical procedures are numerous other fraction separation routines, all of which eventually end up with a (relatively) polar fraction containing the GDGTs. Polar fractions are most commonly stored dried and refrigerated. However, dry storage of dried-down fractions, with unperforated caps at room temperature for up to 20 years has been shown not to affect the outcomes of GDGT analyses (e.g., Sluijs et al., 2020).

Notably underivatized fatty acids will be lost when using an activated alumina column. A silica column can also absorb fatty acids if they are not derivatized first. If these compounds are of interest it is advised to derivatise the TLEs using diazomethane, the less hazardous trimethylsilyldiazomethane or boron trifluoride methanol to convert free fatty acids into methyl esters prior to column separation. Since this paper is on GDGTs we refer to the literature on fatty acids for more information.

## 3.6 Standards

To enable quantification of GDGTs, a synthetic C<sub>46</sub> glycerol trialkyl glycerol tetraether (GTGT) is commonly used as an internal standard (Huguet et al., 2006; see 5.3 for further explanation), although quantification is not required for the calculation of TEX<sub>86</sub> and other GDGT-indices. The C<sub>46</sub> GTGT, or any other standard, can be added in known amount, either (i) to the sediment before the lipid extraction (e.g., Ceccopieri et al., 2019), (ii) to the total lipid extract (Huguet et al., 2006), or (iii) just before the analysis of the polar fraction obtained after column separation. Since lipids can be lost during the extraction process, it is important to be explicit in the methodology section about when the standard is added, and to keep this consistent throughout the workup for all samples in a dataset. This ensures clarity and reproducibility in the experimental procedures (see also in Section 5.3 on quantification).

#### 3.7 Filtering of the polar fraction

Not all polar lipids that eluted from the column using DCM:MeOH dissolve in the solvent mixture that is used during UHPLC-MS analysis (most commonly *n*-hexane-isopropanol 99:1 v/v). In order to remove any particulates before analysis, the polar fraction obtained after column separation and containing the GDGTs is generally filtered through a 0.45 μm pore size PTFE (polytetrafluoroethylene) filter prior to injection (Huguet et al., 2006). This is done using the solvent mixture used during HPLC analysis, so that after filtration the sample is ready to be measured. Partial dry-down to concentrate the samples is to be avoided at this stage, as it possibly affects the solvent mixture composition of the sample, and thus the elution. Complete dry-down and then resuspension in new solvent has the risk of inducing extra particles, which would the require renewed filtering, and a loss of lipids in that process. Brief ultrasonication may redissolve particles. In short, filtering is recommended with GDGTs in the required concentration for HPLC analysis. The optimal concentration of the polar fraction for the UHPLC-MS is about 1 mg mL<sup>-1</sup>. Alternatively, the redissolved polar fraction can be centrifuged before analysis (Coffinet et al., 2014).

## 3.8 Contamination during sample processing

In order to minimize contamination, all metal tools, extraction cells, glassware, aluminum foil, aluminum oxide, silica, Na<sub>2</sub>SO<sub>4</sub>, and glass wool must be furnaced at 450 °C for 2 to 6 hours. If not furnaced, glassware and metal tools need to be cleaned, dried and rinsed (3 times) with solvents before use.

Another potential contamination could come from non-pure solvents. The workup procedure for GDGT analyses requires occasional dry-down of solvents, which has the potential to concentrate contaminants. Routine checks of batches of solvents are therefore key to ensure that solvents are not contaminated. Procedural blank samples should be added to each sample batch to confirm the absence of laboratory contamination. We recommend implementing a blank at the beginning of the laboratory workflow; at lipid extraction (Soxhlet, microwave, accelerated solvent extractor, ultrasonic extraction, see Section 3.3) and at column separation, where different fractions are separated.

Another issue to address during sample processing is the carry-over of GDGTs from one sample to another, either in preparation (powdering) or in the various instruments that are used for extraction. For ASE, inadequate post-extraction cleaning of cells could introduce carry-over contamination, particularly when high-TOC samples are followed by low-TOC samples. Moreover, the system itself could induce carry-over contamination through the tubing that is used to transport the extract from cell to vial (see Sections 2.2 and 3.3.2). Routine measurements of blanks and strict adherence of cleaning protocols minimizes risks of carry-over. For instance, cleaning the ASE cells with MeOH or other polar solvents between sample batches is recommended to minimize the risk of carry-over contamination.

#### 4 LC-MS analysis






Following sample preparation (see Section 3), the filtered polar fractions containing GDGTs are analyzed by liquid chromatography mass spectrometry (LC-MS). High performance liquid chromatography (HPLC) or ultrahigh performance liquid chromatography (UHPLC) systems are commonly used to separate GDGTs. Following compound separation, single, tandem, and high-resolution mass spectrometers are suitable for the detection, identification, and quantification of different compounds (Hopmans et al., 2000; Liu et al., 2012a; Lengger et al., 2018). Round-robin interlaboratory comparison studies (Escala et al., 2009; Schouten et al., 2009; 2013b; De Jonge et al., 2024) have investigated the effects of sample preparation and analytical differences across various LC-MS instruments available in different laboratories worldwide. These studies found no apparent systematic impacts on GDGT-based indices, despite differing extraction and analysis protocols. However, minor discrepancies in absolute GDGT concentrations were noted due to several factors such as instrumentation settings and human error (i.e., manual integration).

The most commonly applied method for the analysis of GDGTs in marine sediments is reported in Hopmans et al. (2016), sometimes with minor modifications. In brief, the separation of GDGTs is achieved using two UHPLC silica columns (Acquity BEH HILIC columns, 2.1 × 150 mm, 1.7 μm; Waters) that are fitted in tandem and use a guard column of the same material (Acquity BEH HILIC pre-column,  $2.1 \times 5$  mm; Waters). Normal phase separation of GDGTs is based on mixtures of n-hexane and isopropanol. The mobile phase is typically composed of mixtures of solvent A, which is 100% n-hexane, with solvent 490 mixture B comprising n-hexane/isopropanol (9:1, v/v). Typically, GDGTs are eluted isocratically for 25 min with 18% B, followed by a linear gradient to 35% B in 25 min, then a linear gradient to 100% B in 30 min. The flow rate is low with 0.2 ml/min and the column temperature is maintained at 30 °C. The typical runtime is 90 min but might be adjusted depending on user requirements (e.g., maximizing peak resolution, required target compounds to be determined, throughput efficiency). About 20 min should be included at the end of each run to return to the initial solvent mixture prior to injection of the next 495 sample, to prevent contamination and to ensure equilibration of the composition of the mobile phase. It was shown that injection volume should not be too high (max 50 µL) when n-hexane-isopropanol (99:1, v/v) is the sample solvent (Wang et al., 2022). Any remaining polar fraction after the LC-MS analysis should be dried under N<sub>2</sub>, properly labeled and stored.

Overall, this method achieves the separation of isoGDGTs, which elute first, followed by branched GDGTs (brGDGTs) and then hydroxylated GDGTs (OH-GDGTs; see Fig. 3). The maximum operating pressure for these columns is approximately 600 bar, but analysis is usually undertaken at much lower operating pressures, commonly ramping up from ca. 180 to 220 bar. The method described by Hopmans et al. (2016) has largely replaced previous approaches using cyano (CN) columns which did not achieve the same degree of separation of several isomers (Hopmans et al., 2000; Schouten et al., 2013b). However, since this development does require some investment, not all laboratories have yet adopted the double-column technique. Based on the new analytical developments, new indices and new paleoclimate calibrations have been proposed that yield lower analytical errors in temperature reconstructions using GDGT-based proxies (e.g., Hopmans et al., 2016). Therefore, it is recommended to use the newer approaches and proxies.





Following column separation, the eluting compounds are ionised by atmospheric pressure chemical ionisation (APCI) using positive polarity mode, which yields protonated molecular ions ([M+H]<sup>+</sup>) of the target compounds. In single quadrupole systems, the spray chamber is operated with gas and vaporizer temperatures of 200 °C and 400 °C, respectively. The drying gas flow is typically set to 6.0 L min<sup>-1</sup> and the nebula pressure to 60 psig. However, all these settings may differ depending on the instrument used and may require individual modifications. The quadrupole temperature is usually set to 100 °C.

The mass selective detector is either a single quadrupole, tandem or high-resolution mass spectrometers. However, only one-dimensional MS mode is typically used to obtain diagnostic M<sup>+</sup> ions of different GDGT homologues and isomers. The

abundances of GDGTs are typically monitored using selective ion monitoring mode (SIM), compounds are identified by comparing mass spectra and retention times with those in the literature. However, analyses performed with an orbitrap mass spectrometer may use alternative detection strategies than SIM. For isoGDGTs, selected ion fragmentograms are m/z 1302.3 (GDGT-0), 1300.3 (GDGT-1 and -1'), 1298.3 (GDGT-2 and -2'), 1296.3 (GDGT-3 and -3'), 1294.3 (GDGT-4 and -4', not commonly targeted since not in the TEX<sub>86</sub> index) and 1292.3 (crenarchaeol and its stereoisomer - cren') (Hopmans et al., 2016; Schouten et al., 2013a). In some environments, isoGDGT homologues up to isoGDGT-8 (m/z 1286) can be detected, particularly in hydrothermal and extremophilic environments (e.g., Schouten et al., 2013). For brGDGTs, target ions are m/z 1050.0, 1048.0, 1046.0, 1036.0, 1034.05, 1032.0, 1022.0, 1020.0, 1018.0 (De Jonge et al., 2014a; Hopmans et al., 2016), as well as brGMGTs at 1048.0, 1034.0 and 1020.0 (e.g., Baxter et al., 2019; Sluijs et al., 2020; Bijl et al., 2021). OH-isoGDGTs are monitored using their M<sup>+</sup> ions at m/z 1318.3, 1316.3 and 1314.3, and as well as dehydrated ions at m/z 1300.3, 1298.3 and 1296.3 for OH-GDGT-0, -1 and -2 respectively (Liu et al., 2012c; Fietz et al., 2016; Varma et al., 2024b). The C<sub>46</sub> GTGT standard is monitored at m/z 743.8. For all ions, a mass window of 1.0 is generally maintained.







The analysis of the large number and diversity of GDGT derivatives and related ether lipids, such as GDDs, GMGTs, GTGTs, can be analyzed with similar LC-MS methods, commonly adapted from the GDGT method of Hopmans et al. (2016). This method can be shortened or extended dependent on which of these compounds are targeted and mainly differ based on the target ions that need to be recorded for identification and quantification (e.g., Coffinet et al., 2015; Naafs et al., 2018; Baxter et al., 2019; Mitrović et al., 2023; Hingley et al., 2024). For instance, isoGMGTs, if present in sample, are already recorded in the mass fragmentograms of isoGDGTs, which are m/z 1300.3 (H-isoGDGT-0), 1298.3 (H-isoGDGT-1), 1296.3 (H-isoGDGT-1) 2), 1294.3 (H-isoGDGT-3), 1292.3 (H-isoGDGT-4) and elute later than isoGDGTs using the Hopmans et al. (2016) method. For isoprenoid GDDs (isoGDDs), the following mass fragmentograms are used: m/z 1246.3 (isoGDD-0), 1244.3 (isoGDD-1), 1242.3 (isoGDD-2), 1240.3 (isoGDD-3), 1238.3 (isoGDD-4) and 1236.3 (isoGDD-cren), which correspond to a mass difference of 56 relative to the equivalent isoGDGTs. Similarly, for branched GDDs (brGDDs), target ions are m/z 966.0 (brGDD-Ia), 964.0 (brGDD-Ib), 962.0 (brGDD-Ic), 980.0 (brGDD-IIa), 978.0 (brGDD-IIb), 976.0 (brGDD-IIc), 994.0 (brGDD-IIIa), 992.0 (brGDD-IIIb) and 990.0 (brGDD-IIIc). IsoGDDs and brGDDs elute later than isoGDGTs and brGDGTs if samples are analysing as reported in Hopmans et al. (2016). In contrast, up to three different isomers of brGDGTs can be detected in each selected ion fragmentogram using m/z 1020.0, 1034.0 and 1048.0, which elute elute after the brGDGTs. Instrument performance (e.g., solvent purity and aging, leaks, blockages, pump functions and pressure control, etc) should be regularly checked and the use of check tunes to regularly evaluate the mass spectrometer performance is recommended. Blanks (processing blanks in each sample batch and clean solvent injections), laboratory standards and known reference samples should be regularly analyzed as part of sample sequences to ensure precision and accuracy of the obtained results. As an example, routine in-house reference sample (Arabian Sea extract mixed with Rowden soil extract) measurements since 2019 on the same instrument ("LC1") in the GeoLab of Utrecht University, show the variability of TEX<sub>86</sub> (Fig. 2; data in Supplementary Table S2). The reference sample measurements during 'normal' performance of the UHPLC-MS (we removed measurements during maintenance) have a standard deviation that is much smaller than the calibration error, and also smaller than any paleoceanographic signal that is reconstructed (0.005 TEX<sub>86</sub> units and 0.008 BIT index units). The stable signal is the result of finetuning and continuous monitoring of instrument performance. The routine analysis of the reference sample ensures reproducibility and allows monitoring of instrument drift, so that when needed instrument settings can be adjusted. Ideally, laboratories use more than one reference sample for robust calibration of the analytical error, and analyse it in replicate over a short time span. Also, for optimal instrument intercalibration, different labs should use the same reference samples so that instruments are optimally tuned to each other. Note also the absence of difference between integrators in the results (Fig. 2). Integration results between integrators can differ considerably, particularly when integration "etiquette" (i.e., how the peak tails are trimmed) is inconsistent. Integration intercomparisons between analysts is strongly recommended when datasets from different integrators are combined. Combining datasets from different instruments require at least the cross comparison of a few samples on each instrument, to ensure both instruments give comparable results for the same samples. Long-term trends in the standard data are probably the result of tuning changes, performance of the pump system and column replacements. Although Fig. 2 demonstrates that the impact of such adjustments to the equipment on GDGT analyses are small, it is recommended that adjustments to the instrument are carefully logged.




# GDGT reference sample (Arabian Sea + Rowden Soil), GeoLab, Utrecht University

Figure 2: The TEX<sub>86</sub> results of over 250 routine analyses of the internal GDGT reference sample (Arabian Sea + Rowden soil) in the Geolab of Utrecht University, over the past 5 years. Colour represents the name of the integrator of the results. Blue line represents a loess smooth through the data, black horizontal line represents the average result of the internal standard, grey bar represents the 1 sigma of all measurements. The standard deviation in TEX<sub>86</sub> index units equates to about 0.3 °C in mid-range temperatures for all calibrations. Measurements of standards during UHPLC-MS maintenance were deleted from this dataset. For BIT results (see Section 6 for details about BIT), relative abundances of individual GDGTs and their standard deviation, see data in Supplementary Table S2.

#### 5 GDGT peak integration

#### 5.1 Integration guidelines




Integrating GDGT peak areas is the first step in interpreting GDGT data. Here, we provide an integration guide, with examples from seafloor surface s in basins with both warm (above 26 °C, typical for tropical and subtropical regions) and cold (below

10 °C, typical for polar and subpolar regions) surface waters, to illustrate temperature-dependent differences in chromatograms and their impact on integration (Figure 3). Traditionally, GDGT-0 through to GDGT-3, crenarchaeol and cren', along with at least the three (five if the 6-methyl brGDGTs is included separately) brGDGTs used in the calculation of the BIT index (see Section 6.1), have been reported for marine samples (see Section 8.2). The utility of a wider array of GDGT-like compounds, including an extended suite of brGDGTs, OH-GDGTs, GMGTs, GTGTs (see Section 8 for details) is increasingly recognized, and we recommend that these are reported where possible, and that it is explicitly stated if not possible (i.e., where abundances of compounds are below detection limit). Figure 3 also demonstrates integrations for GDGT-4 and OH-GDGTs.



Multiple software packages are currently available for automate peak integration (e.g., Dillon and Huang, 2015; Fleming and Tierney, 2016). The pilot results of these programs are impressively close to human integration. These automated approaches can systematize objective choices for the baseline and tail cutoff, can save considerable amount of time when handling large datasets, and reduce the risk of human error in data integration and transfer. However, it is important to note that the inspection of chromatograms for potential coelutions and the verification of proper GDGT concentrations should still be performed manually by an expert.

Figure 3: Example chromatogram and integration guide for two modern samples analyzed using the method described by Hopmans et al. (2016): a) a high sea surface temperature site (mean annual SST ~26.5°C) in the Coral Sea, and b) a low sea surface temperature site (mean annual SST ~-1°C) in the Ross Sea. a) and b) display the total ion current (TIC) for each sample. The mass chromatograms (derived from selective ion monitoring (SIM)) for the six commonly integrated isoGDGTs, as well as GDGT-4 are displayed for the site with warm surface water in c) and for the site with cold surface water in d) further discussion the (note on integration of GDGT-4 in Section 8.3). Mass chromatograms for the three integrated OH-GDGTs are displayed for the site with warm surface water in e) and for the site with cold surface water in f). Grey shaded panels in c), d), e) and f) represent the recommended integration for each GDGT. g) Zoom-in on the baseline of 3 isoGDGTs showing how the tails of these peaks should be trimmed.

#### 5.2 Possible challenges in peak integration

In addition to temperature, the distributions of GDGTs (and likely also GMGTs) can be impacted by spatial variations in archaeal communities, occasionally leading to unusual or challenging peaks to integrate. An example of this is GDGT-2, where an isomer of GDGT-2 can precede and partially coelute with the main isomer peak (Fig. 3d and 3g), leading to a shoulder or double peak. This can be seen in the mass chromatogram of the low SST sample in Figure 3. We recommend that this peak/shoulder should be excluded from integration of the main isomer peak and be noted in the methods section.

Several criteria are in use to determine the detection limit of GDGTs, e.g., below (or above) which quantification of the GDGT is no longer reliable. On the lower end, a signal-to-background noise ratio of 3 is commonly applied, while other laboratories use a minimum peak area cutoff (10³). On the higher end, cutoffs are less well-defined, but in general samples with peaks with 'blunt' maxima are to be diluted and rerun. These values will vary between LC systems and software we recommend that integration limits are stated in publications, and to refer to a peak as NQ, standing for non-quantifiable, to indicate that the peak may still be present but is not abundant enough to be confidently integrated.

#### 5.3 Quantification of GDGTs




GDGTs can be quantified by comparing their peak areas to those of a standard (Huguet et al., 2006; see also Section 3.6). Typically, this is expressed in ng/g, by comparing either to the dry weight of extracted sediment (ng/g dry weight) or to TOC (ng/g TOC) data. Quantification of GDGTs helps in identifying shifts in GDGT concentration and preservation, which is important given that preservation changes (notably post-depositional oxygen exposure) could qualitatively affect the GDGT results (Ding et al., 2013). Although a linear response of the mass spectrometer for all GDGT compounds for quantification is generally assumed, studies on the reproducibility of TEX<sub>86</sub> and BIT index values obtained from GDGT analysis on different MS systems (Escala et al., 2009) and different laboratories (Schouten et al., 2009, 2013b) reveal that this response varies substantially with MS settings. Further, compound quantification using C<sub>46</sub> or any non-GDGT standard will be semi-quantitative, as the response between C<sub>46</sub> and GDGT can vary over time and between instruments. This leads to large interlaboratory offsets particularly in the BIT index, as this proxy comprises compounds over a large *m/z* range (Table 1) (Schouten et al., 2009). Similarly, laboratory-specific MS settings are assumed to cause large (several orders of magnitude) differences in the absolute quantification of GDGTs between laboratories worldwide (De Jonge et al., 2024). These interlaboratory comparison studies have called for the introduction of a community-wide standard mixture with established GDGT proxy values that can be used to calibrate MS instruments (Escala et al. 2009; Schouten et al., 2009, 2013b; De Jonge et al., 2024). Until this has been accomplished, laboratories should perform regular reruns of an in-house GDGT standard

mixture to monitor the stability of their MS instrument, and consistency in integration performance, and thus the accuracy of their measurements (see Section 4). Note, however, that the offsets between instruments in the quantification of GDGTs does not impact individual records generated on the same instrument, i.e., individual instruments, when maintained well, show high/sufficient accuracy and precision in both TEX<sub>86</sub> and BIT estimations (see Section 4).

#### 6 Data interpretation: non-thermal overprints

Along with sea surface temperature, marine sedimentary GDGT distributions are governed by a range of factors that provide insight into past environments but can confound simple interpretation of GDGTs into only one environmental parameter. For example, marine GDGT-based paleothermometry is complicated by a range of sources of isoGDGTs beyond near-surface ammonia-oxidizing Nitrososphaera. Various screening mechanisms and indices (Table 1) have been developed to assess the impact of non-thermal effects or contributions from alternative sources of isoGDGTs. These screening methods typically focus on identifying additional non-surface Nitrososphaera contributions to sedimentary GDGTs, e.g., methanogens or anaerobic methanotrophs (%GDGT-0, methane index; MI) or terrestrial inputs (BIT). The impact of oxic and thermal degradation on GDGT distributions (and associated indices) are not as well developed. However, degradation impacts appear to be minimal, with no evidence that degradation of GDGTs alters their distribution or TEX<sub>86</sub> values during herbivory (Huguet et al., 2006), in the water column (Kim et al., 2009), or in sediments deposited under different redox conditions (Schouten et al., 2004). The most obvious impact of degradation, therefore, is that the pelagic signal is diluted relative to sedimentary archaeal or soil contributions (Huguet et al., 2009). For example, Hou et al. (2023) and Kim and Zhang (2023) showed that high values of the MI, Delta Ring index ( $\Delta RI$ ) and BIT index corresponded with an interval of lower absolute concentrations of GDGTs, and in particular a disproportionate decline in the concentration of crenarchaeol. Because these impacts remain incompletely understood or perhaps offer additional environmental insights, we recommend that samples that are removed by screening should still be reported in the data report but excluded from the subsequent temperature reconstruction. Bijl et al. (2021) provided an R script that follows standardized steps in GDGT data evaluation and interpretation. Although these screening methods have clearly defined cutoff values, we recommend careful consideration of the depositional setting when these screening methods are used. Specifically, concentration changes of GDGTs must be considered for those samples whereby screening methods suggest non-thermal impacts.






Table 1: Summary of screening methods assessing anomalous GDGT distributions.

| Index to identify anomalous                             | Equation                                                                                                                                                                                                                                                                                                                                                                      | Description                                                                                                                                                                                                                                                                                             | Reference                                           |  |
|---------------------------------------------------------|-------------------------------------------------------------------------------------------------------------------------------------------------------------------------------------------------------------------------------------------------------------------------------------------------------------------------------------------------------------------------------|---------------------------------------------------------------------------------------------------------------------------------------------------------------------------------------------------------------------------------------------------------------------------------------------------------|-----------------------------------------------------|--|
| sample                                                  |                                                                                                                                                                                                                                                                                                                                                                               |                                                                                                                                                                                                                                                                                                         |                                                     |  |
| Branched and<br>Isoprenoid<br>tetraether<br>(BIT) index | $BIT = \frac{[brGDGT - I] + [II] + [III]}{[brGDGT - I] + [II] + [III] + [cren]}$                                                                                                                                                                                                                                                                                              | Indicates the relative contribution of terrestrially-derived GDGTs into a marine environment where 0 implies no terrestrial GDGTs and 1 implies no marine contribution. Higher values may indicate a temperature bias in TEX <sub>86</sub> due to terrestrial isoGDGTs.                                 | Hopmans et al. (2004)                               |  |
| Methane Index (MI)                                      | $MI = \frac{[GDGT - 1] + [2] + [3]}{[GDGT - 1] + [2] + [3] + [cren] + [cren']}$                                                                                                                                                                                                                                                                                               | Indicates contribution of methanotrophic Archaea, where higher values (>0.3–0.5) indicate larger contribution.                                                                                                                                                                                          | Zhang et al. (2011)                                 |  |
| %GDGT-0                                                 | $\%[GDGT - 0] = \left(\frac{[GDGT - 0]}{[GDGT - 0] + [cren]}\right) \times 100$ $[GDGT - 0]/[cren]$                                                                                                                                                                                                                                                                           | Indicates contribution of methanogenic Euryarchaeota, where values >67% or ([GDGT-0])/([cren]) >2 indicate larger                                                                                                                                                                                       | Blaga et al. (2009),<br>Sinninghe<br>Damsté et al.  |  |
|                                                         |                                                                                                                                                                                                                                                                                                                                                                               | potential methanogenic input.                                                                                                                                                                                                                                                                           | (2012a)                                             |  |
| Δ Ring Index (ΔRI)                                      | $\Delta RI = RI_{TEX} - RI_{Sample}$ Where: $RI_{TEX} = -0.77 \times TEX_{86} + 3.32 \times (TEX_{86})^2 + 1.59$ And $RI_{Sample} = 0 \times [GDGT - 0] + 1 \times [GDGT - 1] + 2 \times [GDGT - 2] + 3 \times [GDGT - 3] + 4 \times [cren'] + 4 \times [cren']$ And: $TEX_{86}$ $= \frac{[GDGT - 2] + [GDGT - 3] + [cren']}{[GDGT - 1] + [GDGT - 2] + [GDGT - 3] + [cren']}$ | TEX <sub>86</sub> and RI correlate with temperature. ΔRI measures if a sample's GDGT distribution deviates from the modern ocean TEX <sub>86</sub> -RI relationship. If ΔRI lies outside the 95% confidence interval of the modern regression (±0.3 ΔRI units), then non-thermal factors are indicated. | Zhang et al. (2016)                                 |  |
| GDGT-<br>2/GDGT-3                                       | $\frac{[GDGT-2]}{[GDGT-3]}$                                                                                                                                                                                                                                                                                                                                                   | Elevated values (>5) indicate larger contribution from subsurface GDGTs.                                                                                                                                                                                                                                | Taylor et al. (2013) (see also Hurley et al., 2018) |  |
| fcren'                                                  | $f_{cren':cren'+cren} = \frac{[cren']}{[cren] + [cren']}$                                                                                                                                                                                                                                                                                                                     | High values (>0.25) indicate anomalous GDGT distribution impacted by non-thermal factors.                                                                                                                                                                                                               | O'Brien et al. (2017)                               |  |
| %GDGT <sub>RS</sub>                                     | $\%GDGT_{RS} = \left(\frac{[cren']}{[GDGT - 0][+[cren']}\right) \times 100$                                                                                                                                                                                                                                                                                                   | >30% Identifies a 'Red Seatype' distribution, but this cannot be distinguished from a high temperature signal                                                                                                                                                                                           | Inglis et al. (2015)                                |  |

| Dnearest | Distance metric based on a Gaussian Process emulator | High values (>0.5) indicate a    | Dunkley Jones |
|----------|------------------------------------------------------|----------------------------------|---------------|
|          | covariance matrix                                    | GDGT distribution substantially  | et al. (2020) |
|          |                                                      | different from the "nearest      |               |
|          |                                                      | neighbour" within the            |               |
|          |                                                      | calibration data set (i.e., non- |               |
|          |                                                      | analogue to modern surface       |               |
|          |                                                      | sediments)                       |               |

## **6.1 Terrestrial input**




IsoGDGTs form a minor component in terrestrial/aquatic GDGT distributions (e.g., Blaga et al., 2009), but in settings with significant soil organic matter input into the marine realm, terrestrially-derived isoGDGTs have been shown to bias reconstructed temperatures based on TEX<sub>86</sub> values (Weijers et al., 2006a). The Branched and Isoprenoid Tetraether (BIT) index was developed to assess the contribution of GDGTs from soils transported by rivers into marine sediments (Hopmans et al., 2004). It is based on the abundance of the three dominant brGDGTs (brGDGT-I, brGDGT-II, brGDGT-III) compared to crenarchaeol, which is predominantly produced in marine settings (Hopmans et al., 2004). An index value of 0 implies no brGDGT input, while a value of 1 represents no crenarchaeol input. The BIT index is useful in marginal marine and lake sediments (Hopmans et al., 2004). A study on the marine surface sediments of the Congo Fan found that in samples where BIT exceeded 0.4, temperature estimates were biased by >2 °C, implying that terrestrial isoGDGT contributions affect the marine sedimentary isoGDGT pool (Weijers et al., 2006a).

There are however many complications with the use of the BIT index as indicator for terrestrial input. Firstly, BIT index values are not consistent between laboratories (see Section 5.3; Schouten et al., 2009). Secondly, the application of a threshold above which temperature bias is likely to occur is locality dependent because 1) it is influenced by the difference between the TEX<sub>86</sub> values of terrestrially sourced isoGDGTs and marine sourced GDGTs, and 2) the abundance of crenarchaeol, which is typically more abundant at higher temperature, also influences the BIT index (Schouten et al., 2013a). Albeit typically in small amounts, marine production of brGDGTs can also impact BIT index values. However, due to the substantially distinct relative composition of marine-produced brGDGTs from those produced in soils, contributions of marine-sourced brGDGTs can be identified (Peterse et al., 2009; Sinninghe Damsté, 2016). Various indices have been developed to assess whether brGDGTs were terrestrially-sourced or marine-produced (see Section 8.2) (Huguet et al., 2008; Weijers et al., 2014; Xiao et al., 2016). Furthermore, brGDGTs tends to be preferentially preserved over isoGDGTs during syn- and/or post-sedimentary oxic degradation (Huguet et al., 2009). As a compromise, some studies chose to discard TEX<sub>86</sub> as SST proxy when BIT and TEX<sub>86</sub> values correlate, and a location-specific threshold has been established if a correlation existed, or if substantial scatter or anomalous values in TEX<sub>86</sub> occurs above a certain BIT threshold (e.g., Schouten et al., 2009, 2013b; Bijl et al., 2013; Davtian

et al., 2019). Other studies question the use of the BIT index for estimating marine vs. terrestrially derived organic matter (e.g., Walsh et al., 2008), as samples from an increasing range of environments are found to have *in situ* produced brGDGTs (e.g., Peterse et al., 2009; Dearing Crampton-Flood et al., 2019; Bijl et al., 2021). Fietz et al. (2011) argued that the construct of the BIT index makes it also susceptible for changes in crenarchaeol concentrations. Studies in continental shelf areas close to large river systems have also shown that the BIT index decreases rapidly offshore and that river-discharged brGDGTs are not transported far into the marine system (e.g., Sparkes et al., 2015; Yedema et al., 2023). With the documentation of isoGDGT contributions from land, and the production of brGDGTs in the marine system, the BIT index proxy as purely indicative of terrestrial organic matter input becomes problematic. We recommend caution in overinterpreting BIT as proxy for terrestrial input and as criterion for evaluating TEX<sub>86</sub>, particularly when brGDGT distributions diverge from those of modern soils and peats. This means that some of the older TEX<sub>86</sub> data that were discarded because of high BIT might actually accurately reflect local temperature.

## 6.2 Methanogenic input





Marine and sedimentary archaeal communities can also contain methanogenic *Euryarchaeota*, which can synthesize GDGT-0, and likely less so GDGT-1, GDGT-2, and GDGT-3 (e.g., Pancost et al., 2001; Blaga et al., 2009; Sinninghe Damsté et al., 2012a; Inglis et al., 2015). The impact of *Euryarchaeota* on a GDGT distribution can be assessed using %GDGT-0 (Table 1), where values >67% indicate that a sample contains a substantial contribution from methanogenic sourced GDGTs (Sinninghe Damsté et al., 2012a). The ratio of GDGT-0 to crenarchaeol is also sometimes used, with values above 2 indicating a methanogenic source of GDGT-0 (e.g., Blaga et al., 2009; Naeher et al., 2012, 2014b). Other indicators of methanogenic archaeol contributions include archaeol, hydroxyarchaeol, pentamethylicosenes and crocetene (and their derivatives), which can be detected using GC-MS analysis (Hinrichs et al., 2000; Niemann and Elvert, 2008; Naeher et al., 2014b). The impact of methanogenic input on a sedimentary GDGT distribution likely varies by depositional and oceanographic setting, and has typically only been found to have a minor impact in marine sediments (e.g., Inglis et al., 2015; O'Brien et al., 2017). In any case, the likelihood of a methanogenic overprint in a specific oceanographic setting must be assessed when screening to detect non-thermal overprints in GDGTs.

#### 6.3 Anaerobic methanotrophic input

Post-depositional production of isoGDGTs also occurs during anaerobic oxidation of methane by anaerobic methanotrophic (ANME) archaea (Zhang et al., 2011). Methanotrophic archaea, especially those of group ANME-1, preferentially produce GDGT-0, GDGT-1, GDGT-2 and GDGT-3, and can bias reconstructed temperatures in sediments where they are active, such as near cold seeps or regions with presence gas hydrates (Pancost et al., 2001; Zhang et al., 2011). The methane index (MI;

Table 1) assesses the relative contribution of methanotroph-produced GDGTs to those produced in the water column by nonmethanotrophic Nitrososphaera (Zhang et al., 2011). A range of >0.3-0.5 is considered to indicate a significant contribution from a source other than normal marine production (Zhang et al., 2011; Kim and Zhang, 2023). However, we recommend that samples with MI values >0.3 should be treated with caution when estimating SST, acknowledging that similar to the BIT (see above), the MI value may be locality dependent and should be used as a guideline rather than a firm cut off (e.g., Ho et al., 2025; Keller et al., 2025). The sensitivity of the MI has been debated, as it was previously unclear whether small amounts of diagenetic methane in porewater could impact MI values, or if a high flux of methane associated with gas hydrates was required to elevate MI levels (Weijers et al., 2011; Kim and Zhang, 2023). Recent research indicates that the MI is quantitatively related to sedimentary methane diffusive flux, with high MI values strongly associated with high methane fluxes and shallow depths of the sulfate-methane transition zone, where the activity of anaerobic methane oxidation is mostly concentrated (Kim and Zhang, 2023). MI values at the lower end of the range (i.e., >0.3) appear to be associated with high methane flux in polar settings, while this value is closer to >0.5 in non-polar regions (Kim and Zhang, 2023). Additional indicators of methanotrophic archaea include archaeal lipids described in Section 6.2, which are distinguished from methanogenic sources by their <sup>13</sup>Cdepleted composition ( $\delta^{13}$ C values ranging from -40 to low as -120%; e.g., Hinrichs et al., 2000; Niemann and Elvert, 2008; Nacher et al., 2014b). Furthermore, (aerobic) methanotrophic bacteria commonly occur in conjunction with (anaerobic) methanotrophic archaea and are distinguished from heterotrophic bacteria by <sup>13</sup>C-depleted signature of bacterial biomarkers such as fatty acids and hopanoids (Hinrichs et al., 2003; Birgel and Peckmann, 2008; Naeher et al., 2014b). Therefore, the MI may not be an effective indicator at sites with additional sources of GDGTs, such as soil-derived GDGTs in coastal settings (Zhang et al., 2011). Therefore, in conjunction with high MI values,  $\delta^{13}$ C measurements of archaeal and bacterial biomarkers, such as crocetane/archaeol (Kim and Zhang, 2022; 2023) and hopanes (Pancost, 2024), or, in a more elaborate workup scheme, directly on the GDGTs (Pearson et al., 2016; Keller et al., 2025) could be used to assess methanotrophic contributions on the sedimentary GDGT compositions.

#### 6.4 Ring index and Δ Ring Index







Different strains of archaea have been found to display variable TEX<sub>86</sub> values, despite having been cultured at the same temperatures (Elling et al., 2015; Qin et al., 2015). This suggests that growth temperature is not the sole control on changes in the TEX<sub>86</sub> ratio, and that other factors including *Nitrososphaera* community composition can play an important role. A more linear relationship was found between growth temperature and the Ring Index (RI, Table 1), which measures the weighted average of cyclopentyl moieties, across all strains of archaea in culture experiments (Elling et al., 2015; Qin et al., 2015). The slope and strength of this relationship varies between *Nitrososphaera* strains, suggesting that community composition can still impact the relationship between RI and temperature (Elling et al., 2015). Higher values of RI indicate higher temperatures. In the modern ocean, TEX<sub>86</sub> and RI are correlated, and RI can be calculated from TEX<sub>86</sub> using a regression (Zhang et al., 2016).

If a sample's RI deviates from the calculated RI outside of the 95% confidence interval of the modern regression (±0.3 ΔRI units), then the TEX<sub>86</sub> value for that sample is considered to be potentially influenced by non-thermal factors and/or deviates from modern analogues. These factors include the impact of GDGTs derived from soil, methanogenic and methanotrophic archaea as described above, variations in community composition, or potentially other non-thermal impacts on GDGT biosynthesis such as archaeal growth rates (Zhang et al., 2016).

## 6.5 Surface vs subsurface GDGT production (GDGT-2/GDGT-3 ratio)


- One of the known complications of the GDGT-based temperature proxy relates to the depth of GDGT production and export. Nitrososphaera live throughout the water column, are most abundant near the nutricline, and less abundant in the uppermost ~50 meters of the water column (Wuchter et al., 2005; Hurley et al., 2018). Sedimentary GDGTs have typically been considered to best represent surface or shallow subsurface conditions (Schouten et al., 2002), possibly due to preferred export of Nitrososphaera from these depths in aggregates like fecal pellets, as speculated in Wuchter et al. (2005). This was suggested based on the observation that sedimentary TEX<sub>86</sub> values statistically best fit surface temperature (Kim et al., 2008; although below we explain that this might be a statistical artefact caused by the fact the total temperature range is larger in the surface than in the subsurface), and that those sedimentary relationships are only slightly offset from those derived from SPM of surface water (<100 m) (Wuchter et al., 2005; Schouten et al., 2013a; Taylor et al., 2013).
- The GDGT literature suffers from imprecise definition of the qualitative terms surface and (shallow) subsurface. Sediment trap work has indicated that most GDGTs are exported from the surface (e.g., Wuchter et al., 2005). However, in such studies the uppermost trap is typically located at 500 meters below sea surface, proving nothing more than dominant export from the upper 500 meters of the water column, which includes mixed layer and thermocline. For proxy calibration and application, the major factor of importance is whether GDGTs are exported from above, within or below the (permanent) thermocline. Because in many ocean regions the thermocline and nutriclines are related, we might expect that a portion of GDGT export might occur from close to the thermocline rather than only from above.
  - Interestingly, two dominant clades of *Nitrososphaera* are present in the water column, of which one is present in the upper ~200 of the water column (shallow clade), and other resides typically deeper than 1000 m (Francis et al., 2005; Villanueva et al., 2015). The GDGT distribution in the membranes of the shallow clade adjusts to temperature as described by the TEX<sub>86</sub> index, but GDGTs from below 1000 m water depth do not behave in the same way (Schouten et al., 2002; Wuchter et al., 2005; Turich et al., 2007; Taylor et al., 2013; Zhu et al., 2016; Hurley et al., 2018). Contributions of GDGTs derived from the deeper clade can thus affect the temperature signal preserved in marine sediments. The two clades are distinctive in their GDGT-2/GDGT-3 ratio: in SPM collected from above the permanent pycnocline this ratio is typically <5 (e.g., Hernández-Sánchez

et al., 2014; Hurley et al., 2018), while deeper SPM has values up to 40. The exact GDGT-2/GDGT-3 value of deep clade *Nitrososphaera* remains elusive, however, as SPM also includes organic matter exported from the surface ocean. Nonetheless, the essentially bimodal distinction between the two clades, also encountered in the water depth domain, implies that GDGT-2/GDGT-3 ratio values can potentially be used to differentiate between contribution from 'shallow' (~0-200 m depth) and 'deep' (> ~1000 m depth) clades of archaea (Taylor et al., 2013; Kim et al., 2015; Hurley et al., 2018; Rattanasriampaipong et al., 2022; van der Weijst et al., 2022). This implies that calibration of TEX<sub>86</sub> to surface ocean temperature has likely led to an overestimation of the proxy response slope and that integrated (shallow) subsurface calibrations are more appropriate (Ho and Laepple, 2015) Yet, in many oceanographic settings, there is a strong correlation between SST and subT variability (e.g., Ho and Laepple, 2015) which implies that TEX<sub>86</sub> – calibrated to an integrated shallow subsurface depth – may still serve as a proxy for surface temperature variability, if this assumption can be substantiated (Fokkema et al., 2024).

## 6.6 Other non-thermal overprints (f<sub>cren</sub> and Red Sea-type)





Crenarchaeol and cren' are produced by marine *Nitrososphaera*. Some studies (Sinninghe Damsté et al., 2012a; O'Brien et al., 2017) suggested that a substantial increase in the proportion of the cren' relative to crenarchaeol, referred to as f<sub>cren</sub>', could indicate a non-thermal impact on a GDGT distribution. For instance, Group I.1a *Nitrososphaera* produce much less cren' (compared to crenarchaeol) than Group I.1b *Nitrososphaera* (Pitcher et al., 2010), which means that community changes in *Nitrososphaera* will have impact on the GDGT distributions (and thus overprint values and TEX<sub>86</sub>) irrespective of temperature and overprint index values. In the modern core-top dataset, f<sub>cren'</sub> values vary between 0 and 0.16. If a sample displays higher values (i.e., >0.25) it could indicate shifts in archaeal communities. This ratio might be a useful indicator of non-thermal overprints in settings which are warmer than the basins with the warmest sea temperature today. In cultures, f<sub>cren'</sub> variations are strongly temperature controlled, but archaeal ecology might also play a role (Bale et al., 2019).

Core-top samples collected from the modern Red Sea display unusual GDGT distributions and TEX<sub>86</sub> values that do not align with observed temperatures, potentially caused by an endemic *Nitrososphaera* clade (Trommer et al., 2009). Distributions in this highly saline, warm, and low nutrient environment are typically characterized by a low abundance of GDGT-0 relative to the cren' leading Inglis et al. (2015) to suggest that 'Red Sea-type' distributions in the geological past could be detected by the abundance of cren' relative to that of GDGT-0 (%GDGT<sub>RS</sub>). Values of %GDGT<sub>RS</sub> >30 could indicate 'Red Sea-type' distributions, although Inglis et al. (2015) note that these values are also expected to also occur under very high temperature (>30 °C) surface waters. In the Red Sea surface samples, where ecological factors are at play, f<sub>cren'</sub> is high as well, but the fact that the basin water is also hot makes it impossible to separate an ecology effect from a temperature effect.

#### 6.7 D<sub>nearest</sub>

The OPTiMAL, machine learning, approach to GDGT paleothermometry (Dunkley Jones et al., 2020; Table 1) uses a Gaussian process (GP) emulator to determine the relationship between sea surface temperature and all six of the main isoGDGTs (GDGT-0, -1, -2, -3, cren and cren') in the modern calibration dataset. A key motivation of this study was to meaningfully quantify the similarity between any given fossil GDGT assemblage and the modern surface sediment data. OPTiMAL quantifies the distance between the fossil isoGDGT assemblage and its nearest neighbour within the modern calibration data - D<sub>nearest</sub> - using the weighting coefficients on each isoGDGT component – effectively the 'temperature sensitivity' of each component - that have been learned by the GP emulator. A threshold D<sub>nearest</sub> value of 0.5 was proposed at the inflection point of rapidly increasing uncertainty in the GP (Dunkley Jones et al., 2020). With D<sub>nearest</sub> >0.5 fossil samples are considered to be significantly 'non-analogue' relative to the modern, and OPTiMAL has limited or no predictive power for SST estimation. Irrespective of OPTiMAL as an SST predictor, D<sub>nearest</sub> can be used to detect non-analogue fossil isoGDGT assemblages in the components that are relevant to SST reconstruction.

## 7 Temperature calibrations




#### 7.1 GDGT surface sediment dataset

GDGT distributions in samples from a global surface sediment dataset have been a central asset in developing TEX<sub>86</sub> as reliable paleotemperature proxy. The calibration dataset was initially composed of 44 surface sediment samples (Schouten et al., 2002) and the dataset has been expanded several times since then (Kim et al., 2008, 2010), to the most recent iteration of 1095 samples in Tierney and Tingley (2015). More surface sediment data points/sites have been reported since 2015, in some cases substantially expanding some areas of the global ocean that are underrepresented in the data set of Tierney and Tingley (2015), for example from the Mediterranean Sea, Southern Ocean and Antarctica (e.g., Kim et al., 2015; Jaeschke et al., 2017; Lamping et al., 2021; Harning et al., 2023) (Fig. 1). However, despite these efforts, several regions of the global ocean remain poorly represented: subtropical gyres, deep water settings and other areas of the open ocean distal from land (Fig. 1). On top of this, surface sediment data were assembled while the proxy was still in the development phase. This means that first surface sediment datasets included the six primary isoGDGTs (GDGT-0, GDGT-1. GDGT-2, GDGT-3, crenarchaeol and cren') and the three brGDGTs included in the BIT index (brGDGT-I, brGDGT-II and brGDGT-III), and did not exclude samples with non-thermal overprints. Also, it misses more recent developments in the proxies, e.g., in the interpretations of brGDGTs that are marine-*in situ* produced, and the inclusion of GDGT-4. Following our recommendations in Section 10 and in Table 4, we envision that an open access repository in which all cope top data are easy accessible and stored with metadata, will enable the community to iteratively improve the calibrations.

GDGTs are commonly applied to paleo/geological samples, where the following information/data are typically generated e.g.,: sediment fraction (clay, silt), mineralogical composition, carbonate content, X-ray Fluorescence (XRF), TOC, or/and depositional setting (shallow marine, hemipelagic, etc.). To ensure better understanding of GDGT distributions in ancient and recent samples, future expansion of the surface sediment data set as well as critical evaluation of the existing surface sediment data should include the following information:

- Surface sediment GDGT data should be reported as peak areas, with quantified concentrations where available, as well as the used detection limit (e.g., signal-to-noise ratio, peak area, etc). We also suggest expanding the range of GDGTs beyond those used for TEX<sub>86</sub> (see Section 8). This will optimize the interoperability and reusability of surface sediment data as different indices or calibration methods are developed over time, that include more than the six primary isoGDGTs applied in the first top core calibration in of Schouten et al. in 2002.
- Samples displaying unusual distributions (i.e., that fail the screening methods described above) should still be reported but flagged as possibly impacted by non-thermal processes. As an example of how this could be done, the R-script of Bijl et al. (2021) indicates in columns with logical values which samples show unusual distributions based on threshold values for overprint criteria.
- Currently sea surface temperature and water depth to seafloor are reported for each surface sediment site. We
  recommend that a larger range of environmental metadata is included, for example, distance to shore, water column
  structure/water mass, oxygen concentrations, and any information that can guide an interpretation of surface sediment
  age.
- Core-top and surface sediment is often assumed to be representative of modern conditions, but variability in sediment dynamics, bottom currents and even sampling methods may lead to these samples representing thousands of years of sediment accumulation (Mekik and Anderson, 2018). Where possible, an estimate of sediment age (i.e., based on microfossils or <sup>210</sup>Pb dating) should be reported.
- Surface sediment data will need to be scrutinized regarding data quality and confounding factors to arrive at proper proxy calibrations. For example, there may be some doubt regarding data quality of some samples included in the current dataset that must be overcome by data reproduction in different laboratories (i.e., round-robins). The community should reach concensus on a way forward for defining analytical conditions and use of surface sediment data for the development of proxies.

# 860 7.2 Calibration equations






Currently, a large range of GDGT-based temperature calibrations exist, as a result of method development and improved mechanistic understanding of proxy functioning. We summarize key developments in global calibrations, and when/how to

use these calibrations in Table 2. We note that the current understanding of GDGT biosynthesis and taphonomy prevents choosing one calibration that suits all purposes, geologic time intervals, and geographic settings.

Table 2. Summary of key developments in GDGT-based global calibrations (where 'n' refers to the number of coretop/surface sediment samples, RSE - residual standard error).

| Calibration                                   | Equation                                                                                                                                                                       | Calibration error                              | Description                                                                                                                                               | Reference               | Status                                                                                                                                                                                                                 |
|-----------------------------------------------|--------------------------------------------------------------------------------------------------------------------------------------------------------------------------------|------------------------------------------------|-----------------------------------------------------------------------------------------------------------------------------------------------------------|-------------------------|------------------------------------------------------------------------------------------------------------------------------------------------------------------------------------------------------------------------|
| Linear TEX <sub>86</sub> ,<br>n=44            | $TEX_{86} = 0.015 \times T + 0.28$ Where $TEX_{86}$ $= \frac{[GDGT - 2] + [GDGT - 3] + [cren']}{[GDGT - 1] + [GDGT - 2] + [GDGT - 3] + [cren']}$ And T= annual mean SST in °C. | RSE ±2.0 °C                                    | Original linear<br>SST calibration<br>based on a global<br>surface sediment<br>dataset                                                                    | Schouten et al. (2002)  | Superseded by Kim et al. (2008)                                                                                                                                                                                        |
| TEX <sub>86</sub> ' n=104                     | TEX <sub>86</sub> '=0.016×SST+0.20  Where TEX <sub>86</sub> '= ([GDGT-2] +[GDGT-3] +[cren'])/ ([GDGT-1] +[GDGT-2] +[cren'])                                                    | RSE: none<br>given                             | Modified version<br>of TEX <sub>86</sub> used in<br>Paleogene Arctic<br>samples with high<br>relative<br>abundances of<br>GDGT-3                          | Sluijs et al.<br>(2006) | No longer in use,<br>see discussion in<br>Sluijs et al. (2020)                                                                                                                                                         |
| Updated linear<br>TEX <sub>86</sub><br>n=223  | $SST = 56.2 \times TEX_{86} - 10.8$                                                                                                                                            | RSE ±1.7 °C                                    | Updated linear<br>SST calibration                                                                                                                         | Kim et al. (2008)       | Superseded by Kim et al. (2010)                                                                                                                                                                                        |
| Reciprocal<br>TEX <sub>86</sub><br>n=287      | $SST = -16.3 \times \left(\frac{1}{TEX_{86}}\right) + 50.5$                                                                                                                    | 68.2%<br>confidence<br>interval                | Non-linear<br>(reciprocal), high<br>temperature<br>calibration                                                                                            | Liu et al. (2009)       | Not recommended<br>since it lacks<br>underlying<br>mechanistic<br>understanding                                                                                                                                        |
| TEX# <sub>6</sub><br>Exponential<br>n=255     | $TEX_{86}^{H} = log (TEX_{86})$ And $SST = 68.4 \times (TEX_{86}^{H}) + 38.6$                                                                                                  | TEX#6                                          | exponential (Kim et al., 2010; Tierney and Tingley, 2014); TEX <sup>H</sup> <sub>86</sub> was recommended for temperature > 15 °C. Surface waters (0–20m) | Kim et al. (2010)       | In use, but TEX <sub>86</sub> suffers from statistical shortcomings, notably regression dilution and residuals at the warm end of the calibration. (Tierney and Tingly, 2014)                                          |
| TEX <sub>86</sub> and<br>Exponential<br>n=396 | $TEX_{86}^{L} = log \left( \frac{[GDGT - 2]}{[GDGT - 1] + [GDGT - 2] + [GDGT - 3]} \right)$ And $SST = 67.5 \times (TEX_{86}^{L}) + 46.9$                                      | RSE $\pm 4.0$ °C for $TEX_{86}^L$ $\pm 2.5$ °C | TEX <sub>86</sub> was recommended for temperature below 15 °C. Surface waters (0–20m)                                                                     | Kim et al.<br>(2010)    | TEX <sub>86</sub> has mechanistic flaws and is no longer recommended for use (e.g., Inglis et al., 2015; Taylor et al., 2018). except in the Baltic Sea (e.g., Kabel et al., 2012) and other polar regions (Ai et al., |

|                                                             |                                                            |                                                                |                                                                                                                                                                                                                                                               |                                           | 2023), where the production of cren' is limited                                                                                                                                                                                                  |
|-------------------------------------------------------------|------------------------------------------------------------|----------------------------------------------------------------|---------------------------------------------------------------------------------------------------------------------------------------------------------------------------------------------------------------------------------------------------------------|-------------------------------------------|--------------------------------------------------------------------------------------------------------------------------------------------------------------------------------------------------------------------------------------------------|
| Linear TEX <sub>86</sub> n=21                               | $T = 52.0 \times (TEX_{86}^H) + 42.0$                      | RSE ±3.4 °C                                                    | Linear calibration of $TEX_{86}^H$ in mesocosms.                                                                                                                                                                                                              | Kim et al. (2010)                         | Not used because of<br>the lack of analogy<br>between mesocosm<br>and surface<br>sediment GDGT<br>distributions                                                                                                                                  |
| BAYSPAR<br>n=1095<br>(samples north<br>of 70° N<br>removed) | Bayesian, spatially varying regression based on $TEX_{86}$ | 90 <sup>th</sup> percent<br>confidence<br>intervals            | Linear calibration. Used in 'Standard' or 'Analogue' mode depending on whether oceanographic conditions at the site were analogue to modern. Can reconstruct SST or SubT (weighted 0– 200m water depth, weights given by gamma probability density function). | Tierney and<br>Tingley<br>(2014,<br>2015) | In use, particularly<br>for applications in<br>modern-like<br>temperature ranges<br>and analogue<br>settings (e.g., Hollis<br>et al., 2019)                                                                                                      |
| Subsurface  TEX# 86  n=255; >15 °C                          | $T = 40.8 \times (TEX_{86}^H) + 22.3$                      | 95%<br>confidence<br>intervals                                 | Recalibration of TEX <sub>86</sub> . A least squared regression was performed for depth-integrated temperatures between 0-1000m water depth, and an ensemble (SUBCAL) derived for a subsurface temperature calibration.                                       | Ho and<br>Laepple<br>(2016)               | In use by the community because the paper presents a useful range of integrated export depths (e.g., van der Weijst et al., 2022). The assumption of deeper export reduces the proxy response slope and dampens variability in downcore records. |
| OPTiMAL                                                     | Machine learning-based Gaussian process regression model   | 95% confidence intervals, root mean square uncertainty ±3.6 °C | SST model based<br>on machine<br>learning using<br>relative<br>abundances of all<br>6 isoGDGTs.                                                                                                                                                               | Dunkley<br>Jones et al.<br>(2020)         | In use, but assumes surface signal. It cannot be used to predict temperatures outside of the modern calibration range of SSTs (>30 °C).                                                                                                          |

An initial calibration between GDGTs and temperature was developed by Schouten et al. (2002) on a surface sediment dataset of 44 samples (Table 2). They found the best correlation with annual mean SST was a linear regression using the TEX<sub>86</sub> ratio,

which includes GDGT-1, GDGT-2, GDGT-3 and cren', but excludes the abundant GDGT-0 and crenarchaeol to avoid these compounds having a large influence on the index. This was also driven by concerns that GDGT-0 had many alternative sedimentary sources (see Section 6). Nonetheless, Zhang et al. (2015) confirmed a strong relationship between SST and the weighted average RI. A modified version of TEX<sub>86</sub>, termed TEX<sub>86</sub>', based on an expanded data set of 104 surface sediment sites, was used by Sluijs et al. (2006) for samples from the Paleogene succession from the Arctic which contained high relative abundances of GDGT-3, possibly related to high terrestrial input. This index removed GDGT-3 from the denominator of TEX<sub>86</sub>, but proportionally high GDGT-3 was not commonly found in other sample sets, and the index is no longer in use (Sluijs et al., 2020).

The linear TEX<sub>86</sub>-based calibration of Schouten et al. (2002) was subsequently updated by Kim et al. (2008) based on an expanded global surface sediment data set of 223 samples (Table 2). Kim et al. (2008) noted that the surface sediment samples from high latitude sites showed significant scatter, and that the TEX<sub>86</sub> calibration had limited utility below 5 °C. Liu et al. (2009) expanded on the concepts introduced by Schouten et al. (2002) acknowledging the challenges in extrapolating the surface sediment calibration above the limit of the modern day (i.e., above ~30 °C and TEX<sub>86</sub> values of ~0.73) (Table 2). Liu et al. (2009) developed a calibration based on the reciprocal of TEX<sub>86</sub> that reduced the slope of the TEX<sub>86</sub>-temperature relationship for samples from warm water pool.

The modern surface sediment data set was expanded further by Kim et al. (2010), who also observed the relative insensitivity of TEX<sub>86</sub> to temperature in cold regions and investigated variations in GDGT ratios to improve the calibration at both the cold and warm ends of the spectrum (Table 2). The authors concluded that an exponential form of TEX<sub>86</sub>, referred to as  $TEX_{86}^H$  was most optimal as it exhibited the highest R<sup>2</sup> and the smallest residual error (which was their prime quality criterion) for samples from sites with moderate to high surface water temperature (15–28 °C) when tested on a surface sediment dataset with subpolar and polar samples removed, and recommended this calibration was applied for sites with expected temperatures above 15 °C. For the full modern temperature range, and especially below 15 °C, the authors found an exponential form of a GDGT ratio without the cren' resulted in the best correlation ( $TEX_{86}^L$ ).  $TEX_{86}^L$  is still in use as a proxy for cold, polar sea surface or subsurface temperature (e.g., Kabel et al., 2012; Ai et al., 2024), where cren' is not abundantly produced, and there it seems to provide the most plausible absolute values. However, beyond polar regions  $TEX_{86}^L$  does not show a linear correlation with an increase in cyclopentane moieties, i.e., it lacks a physiological basis and can be easily biased by water depth-driven variations as expressed by the GDGT-2/GDGT-3 ratio (Taylor et al., 2013). Thus, this calibration of TEX<sub>86</sub> has been widely discarded by the community, and we recommend it not be used outside cold, polar waters.

Kim et al. (2010) also investigated the validity of  $TEX_{86}^H$  and  $TEX_{86}^L$  using mesocosm experiments, and found that  $TEX_{86}^H$  (i.e.,  $log(TEX_{86})$ ) provided the strongest correlation to incubation temperatures in culture data, albeit with a slightly different

intercept and slope reflecting a reduced amount of the cren' in cultures than present in surface sediments. Incubation studies have since highlighted that TEX<sub>86</sub>-temperature correlations vary across archaeal strains, while the RI appears to have a more linear relationship (Elling et al., 2015). Despite this, the RI values for global surface sediments have not yet been calibrated to temperature.

Several temperature calibrations have focused on the fact that sedimentary GDGTs may represent a subsurface rather than near surface temperature signal (e.g., Taylor et al., 2013). Kim et al. (2008) statistically compared the fit of sedimentary TEX<sub>86</sub> to temperatures from various depths, as outlined above, and based on that proposed that TEX<sub>86</sub> correlates best to SST. Later, mounting evidence suggested that TEX<sub>86</sub> is more representative of a subsurface signal. Ho and Laepple (2016) employed a calibration ensemble between 0 and 1000 m water depth, assuming the majority of GDGTs are exported from between 100 and 350 m. Tierney and Tingley (2014, 2015) expanded the surface sediment dataset and developed a Bayesian, spatially varying method to generate linear calibrations (BAYSPAR) to surface (0–20m) or subsurface (0–200m), and recognized a regression dilution bias in the warm end of  $TEX_{86}^H$ . An approach taken by van der Weijst et al. (2022a) combines modern observations of water column structure, and additional microfossil and GDGT-based proxies (i.e., GDGT-2/GDGT-3 ratio) to assess changes in the export depth of GDGTs through a 15 million year long equatorial Atlantic record, enabling authors to determine which depth-integrated calibration is most appropriate to use at in the investigated core site.

BAYSPAR can also spatially weight a calibration to surface sediments near a sample site, recognizing that archaeal communities vary through the global ocean. As well as BAYSPAR, several regional calibrations have been developed to take account of this spatial variance, with examples including a calibration for the Baltic Sea (Kabel et al., 2012) or Sea of Okhotsk (Seki et al., 2014) and a subsurface calibration for offshore Antarctica (Kim et al., 2012). More recently, improvements to low temperature calibrations have focused on the inclusion of OH-GDGTs alongside isoGDGTs (e.g., Fietz et al., 2016; Varma et al., 2024b) (see Section 8.1).

Machine learning-based approaches (e.g., OPTiMAL, Dunkley Jones et al., 2020) take an agnostic view of the form of the relationship between GDGT abundances and temperatures, using a Gaussian Process emulator to optimize temperature estimation from the modern core-top calibration data set of Tierney and Tingley (2015). The disadvantage of this approach is that no SST estimations can be made outside of the range of the calibration space - the GP emulator can only make SST predictions with any degree of confidence where it is constrained by data, so for SSTs >30 °C and <5°C. Moreover, the agnostic approach of OPTiMAL ignores our increasing understanding of proxy functioning, such as spatial variability in the depth of production.

Choosing the most appropriate calibration can be a complicated process, creating at least two particularly acute challenges to their interpretation (e.g., Fokkema et al., 2024). First, although evidence for a dominant subsurface signal is mounting, it remains unclear which water depths sedimentary TEX<sub>86</sub> exactly represents at certain oceanographic conditions. Although several robust calibration approaches calibrate GDGTs to sea surface temperature and show good proxy-to-proxy agreement, the majority of GDGTs are produced in the subsurface (50–200m; Hurley et al., 2018). Simultanously, dominant depth of export remains difficult to constrain, and for deeper time, a substantial part of the TEX<sub>86</sub> proxy records is derived from settings with relatively shallow water depths (Tierney et al., 2017). Proxy records should be interpreted with that in mind; our lack of understanding about how temperature affects GDGT distributions of the Archaea living in sub-thermocline waters and responsible for unusual GDGT-2/GDGT-3 ratios. For samples from the most recent geological past, where oceanographic conditions could be assumed to be relatively similar to modern, investigating how different calibrations predict temperature for nearby core-tops may also help to inform an appropriate calibration to use. Second, although calibration choice may not strongly impact reconstructed temperatures when applied to indices that fall within the modern calibration dataset, the assumed mathematical relationship (linear or logarithmic) between TEX<sub>86</sub> and temperature has a profound impact when applied to ancient climates with indices higher than those observed in modern oceans (e.g., Cramwinckel et al., 2018; Hollis et al., 2019). It is important to note that there is not necessarily one 'correct' calibration to use, as calibration choice will depend on factors such as sample location and the studied time period. It is critical to determine the mathematical relationship between temperature and GDGT distributions, including whether that is properly represented by TEX<sub>86</sub> at all temperatures (i.e., as opposed to a weighted averaged ring index). We must also improve our understanding of the mechanistic relationship between GDGTs and temperature, including the effect of local oceanographic conditions (e.g., ocean stratification, thermocline/nutricline depths), community structure, export dynamics and changes therein. Therefore, when publishing the GDGT-derived temperature record, justification of the selected calibration should be provided. It is also imperative to disclose GDGT data in full for appropriate reuse and recalibration of existing data (see Section 6).







For the moment, there is no perfect calibration that provides reliable temperature reconstructions in cold temperatures, warmer-than-modern climates, and that takes full account of the depth of production of GDGTs, also given the fact that that depth may vary per oceanographic setting and through time. By evaluating known overprints, the depth of production and the oceanographic setting, the community has a way forward for further proxy development, evaluating surface sediment datasets and through that, improving calibrations for more accurate application of paleothermometry.

### 8 Other marine GDGT proxies

#### 8.1 OH-GDGTs







OH-GDGTs – much like isoGDGTs – are widespread in marine environments. They contain one or two hydroxy groups attached to their biphytanyl chains, and were first identified in marine sediments by Liu et al. (2012c). OH-GDGTs increase in abundance at higher latitudes (Huguet et al., 2013). Initially, OH-GDGTs were used to improve accuracy of the SST reconstructions in (sub)polar regions where the TEX<sub>86</sub> residual standard error increases. Using a dataset of 77 samples collected from the water column, marine surface sediments, as well as marine and freshwater downcore sediments, Huguet et al. (2013) found that the relative abundance of OH-GDGTs compared with isoGDGTs (%OH) shows a weak negative correlation with annual SST. They reduced the dataset to marine surface samples only (n = 38; Table 3) in order to improve the correlation and reduce the error. By extending the initial dataset of Huguet et al. (2013) with sea surface sediments from the Southern Ocean (Ho et al., 2014) (n = 52) and empirically searching for a better calibration, Fietz et al. (2016) established a new temperature calibration including GDGT-1, GDGT-2, GDGT-3, cren' and OH-GDGTs (OH<sup>C</sup>) that shows a better correlation and lower residual standard error than %OH with annual SST, summer SST and SST 0 – 200m. Recently, Varma et al. (2024b) compiled OH-GDGT from an extended array of surface sediments, including data analysed at NIOZ (i.e., the 'NIOZ dataset'; n=575) and data analysed in other laboratories (n=297). Data that failed either the screening methods described in Section 6 (i.e., high BIT index values), or where abundances of OH-GDGT-1 and/or OH-GDGT-2 were below the detection limit, were excluded from further analysis, leaving n=469 in the NIOZ dataset. Varma et al. (2024b) found interlaboratory offsets for OH-GDGTbased proxies between the NIOZ dataset and datasets from other laboratories. The offset was especially large for indices which combine both iso- and OH-GDGTs (i.e., %OH), indicating OH-GDGT response may vary on different analytical equipment. To circumvent this, authors obtained calibration results based only on the NIOZ dataset, but suggested that a round robin study is necessary to determine the extent of interlaboratory differences. Authors found a better correlation between annual SST and %OH compared with earlier studies, but with no significant improvement for the OH<sup>C</sup> calibration. Varma et al. (2024b) also proposed a new OH-GDGT-based temperature calibration by adding OH-GDGT-0 to the denominator of the TEX<sub>86</sub> equation (TEX<sub>86</sub><sup>OH</sup>) (Table 3). This new calibration has a stronger correlation with SST than TEX<sub>86</sub>, showing no flattening of the relationship below 15 °C, remaining linear down to around 5 °C. Interestingly, the TEX<sub>86</sub><sup>OH</sup> calibration has a similar correlation with ocean temperatures between 0–200 m ( $R^2 = 0.89$ , n = 470) as with SST ( $R^2 = 0.88$ , n = 513), suggesting that TEX<sub>86</sub>OH might be more suitable for reconstructing subsurface temperatures than surface temperatures, although as of yet it is not known where in the water column OH-GDGTs are produced.

Another approach is to generate OH-GDGT calibrations independent of isoGDGTs, also circumventing the interlaboratory differences for proxy indices that incorporate both OH-GDGTs and isoGDGTs (Varma et al., 2024b). Lü et al. (2015) analyzed

the correlation between OH-GDGT cyclization and SST using a dataset of 107 samples from the global dataset of Huguet et al. (2013), the Nordic Seas (Fietz et al., 2013) and new sediment samples from the South and East China Seas. Lü et al. (2015) proposed two new calibrations: RI-OH using only OH-GDGT-1 and OH-GDGT-2, recommended for SST > 15 °C, and RI-OH' in which OH-0 is added to the denominator of the RI-OH equation, recommended for SST < 15 °C. Subsequently, the correlation of RI-OH' with SST was improved by Fietz et al. (2020) by adding surface sediment data from the Baltic Sea (Kaiser and Arz, 2016), observing a better correlation to SST ( $R^2 = 0.76$ ) than in the original equation ( $R^2 = 0.75$ ) of Lü et al. (2015). Recently, Varma et al. (2024b) updated the equations of these two calibrations, improving the correlation of RI-OH with SST ( $R^2 = 0.79$ ) but showing a poorer correlation for RI-OH' ( $R^2 = 0.64$ ).



995

Although the main factor influence on OH-GDGT distributions is temperature, several studies have shown the impact of non-thermal factors. Xiao et al. (2023) observed that the production of OH-GDGTs by benthic archaea can have a large impact on RI-OH', which dictates caution to apply this proxy on sediments from deep ocean basins. Other confounding factors include: i) the influence of seasonal phenomena such as the extension of sea ice cover (Wu et al., 2020), or changes in the monsoon regime (Wei et al., 2020), ii) inputs of terrestrial sediments that often have a higher relative abundance of OH-GDGT-2 and lower relative abundance of OH-GDGT-1 and %OH compared to marine sediments (Kang et al., 2017; He et al., 2024; Varma et al., 2025), iii) freshwater inputs modifying how the archaea adjust the OH-GDGT composition of their membrane (Sinninghe Damsté et al., 2022), iv) a difference in the archaeal community as a function of water column depth (Zhu et al., 2016; Lü et al., 2019; Liu et al., 2020; Varma et al., 2023, 2024a), v) changes in dissolved oxygen concentration or nutrient abundance (Harning et al., 2023; Harning and Sepúlveda, 2024), vi) seasonal biases (Lü et al., 2015; Davtian et al., 2019), and vii) difficulty in quantifying OH-GDGTs when abundance is low, especially in tropical regions with temperatures > 25 °C (Varma et al., 2024a, 2024b).

As with TEX<sub>86</sub>, selecting a calibration and an equation is a complicated process that is generally carried out on a case-by-case basis, depending on the characteristics of the study area (e.g., ice cover, variability of terrestrial inputs, depth of the nutricline, archaeal community differences) and the location of the samples (e.g., water depth, distance from the river mouth). Importantly, a brief justification of the choice of calibration is necessary to allow the reader to understand the rationale behind this decision.

### 8.2 GDGT-4

Although GDGT-4 is produced by marine *Nitrososphaera*, it is rarely quantified and reported with other GDGTs because it is not included in the TEX<sub>86</sub> equation. However, its presence in various environments, including cultures (Pitcher et al., 2010; Elling et al., 2015, 2017; Bale et al., 2019), SPM (Zhu et al., 2016; Hurley et al., 2018; Besseling et al., 2019), surface sediments (Wei et al., 2011), ancient sediments (Zhang et al., 2014; Zhuang et al., 2017; De Bar et al., 2019; Crouch et al., 2020; Cavalheiro et al., 2021) and hydrothermal systems (Hernández-Sánchez et al., 2024) suggests that GDGT-4 contributes to

membrane adaptation, and may be an important component of membrane lipids, especially in warm climates. Although GDGT-4 is not currently included in GDGT-derived indices, quantifying it and including it in the surface sediment database will future-proof data for potential inclusion in future indices. Accurate quantification of GDGT-4 hinges on full chromatographic separation of crenarchaeol (or any crenarchaeol isomer) and GDGT-4. If crenarchaeol and GDGT-4 are not chromatographically separated, correction of the apparent GDGT-4 peak area is needed, which may be done by subtracting the isobaric interference of the +2 Da isotope peak of crenarchaeol (1294.2601), which occurs at 45.97% of the intensity of crenarchaeol (1292.2444) at natural isotopic abundance (Sinninghe Damsté et al., 2012b).

Table 3. Temperature calibrations using OH-GDGTs (RMSE - root mean square error)

| Calibration     | Equation                                                                                                                                                       | Error                                       | Description                                                                                                         | Reference            |
|-----------------|----------------------------------------------------------------------------------------------------------------------------------------------------------------|---------------------------------------------|---------------------------------------------------------------------------------------------------------------------|----------------------|
| % ОН            | $\frac{\sum[OH - GDGTs]}{\sum[OH - GDGTs] + \sum[iso - GDGTs]}$ $SST = -0.24 \times \%OH + 8.3$                                                                | RMSE = 9.7 °C<br>95% confidence<br>interval | Original global linear SST calibration based on a marine surface sediment dataset of 38 samples.                    | Huguet et al. (2013) |
| RI-OH           | $\frac{[OH - 1] + 2 \times [OH - 2]}{[OH - 1] + [OH - 2]}$ $RI - OH = 0.018 \times SST + 1.11$                                                                 | RMSE = 6 °C                                 | Global linear calibration<br>based on marine surface<br>sediment dataset of 107<br>samples, for use above<br>15 °C. | Lü et al. (2015)     |
| RI-OH′          | $\frac{[OH - 1] + 2 \times [OH - 2]}{[OH - 0] + [OH - 1] + [OH - 2]}$ $RI - OH' = 0.0382 \times SST + 0.1$                                                     | RMSE = 6 °C                                 | Global linear calibration based on marine surface sediment dataset of 107 samples, for use below 15 °C.             | Lü et al. (2015)     |
| $\mathrm{OH^c}$ | $\frac{[GDGT - 2] + [GDGT - 3] + [Cren'] + [OH - 0]}{[GDGT - 1] + [GDGT - 2] + [GDGT - 3] + [Cren'] + \sum [OH - GDGTs]}$ $OH^{C} = 0.0266 \times SST - 0.144$ | RMSE = 3.9 °C                               |                                                                                                                     | Fietz et al. (2016)  |

| ТЕХ <sup>ОН</sup>      | $\frac{[GDGT-2]+[GDGT-3]+[Cren']}{[GDGT-1]+[GDGT-2]+[GDGT-3]+[Cren']+[OH-0]}$ $TEX_{86}^{OH}=0.023\times SST+0.03$ $TEX_{86}^{OH}=0.026\times SST_{0-200m}+0.09$ NIOZ dataset $TEX_{86}^{OH}=0.021\times SST+0.08$ $TEX_{86}^{OH}=0.025\times SST_{0-200m}+0.11$ Complete dataset | Standard deviation of residuals (NIOZ dataset): 3.2 °C (SST) and 2.8 °C (SST 0–200m)  Standard deviation of residuals (complete dataset): 3.7 °C (SST) and 2.9 °C (SST 0–200m) | Increases the temperature sensitivity of the index, especially for temperatures from 5 to 15 °C. Two equations available: surface and subsurface (0–200m). Calibrations obtained using the NIOZ dataset and complete dataset.                                                                                                                                    | Varma et al. (2024b) |
|------------------------|-----------------------------------------------------------------------------------------------------------------------------------------------------------------------------------------------------------------------------------------------------------------------------------|--------------------------------------------------------------------------------------------------------------------------------------------------------------------------------|------------------------------------------------------------------------------------------------------------------------------------------------------------------------------------------------------------------------------------------------------------------------------------------------------------------------------------------------------------------|----------------------|
| RI — OH' <sub>WD</sub> | $SST = 15.2 \times RI - OH'_{200m} + 5.0$ for SST reconstruction between 0 – 200 m $SST = 21.2 \times RI - OH'_{Global-FOD} + 0.0013 \times Water Depth - 0.3$                                                                                                                    | RMSE = $2.1 ^{\circ}$ C                                                                                                                                                        | The contribution of deep-water archaea may alter OH-GDGT distributions, with increased OH-GDGT-0 production at greater depths due to colder temperatures. This effect is more pronounced at low latitudes, where the surface-to-bottom water temperature gradient is stronger. The RI-OH' equations proposed here attempt to consider the impact of water depth. | Xiao et al. (2023)   |

#### 8.3 Branched GDGTs

The distribution of brGDGTs in terrestrial environments is mainly linked to changes in temperature and pH (Weijers et al., 2007b). As brGDGTs are predominantly produced in terrestrial environments, their presence in marine sediments is often associated with the input of terrigenous material, notably soils (Hopmans et al., 2004; Schouten et al., 2013a). However, studies carried out in a variety of environments including open oceans (Weijers et al., 2014), fjords (Peterse et al., 2009), continental shelves and rivers (Zhu et al., 2011; Zell et al., 2014), and the deepest hadal trenches (Xiao et al., 2020) showed differences in the distribution of brGDGTs between terrestrial and marine sediments, leading to the hypothesis of *in situ* marine production. The distribution of *in situ* brGDGTs in marine environments remains however poorly understood. Currently, three approaches are used to differentiate the origin of brGDGTs in marine sediments: the abundance of hexamethylated (sum of hexamethylated brGDGTs, ΣΙΙα) over pentamethylated (sum of pentamethylated brGDGTs, ΣΙΙα) brGDGTs (ΣΙΙΙα/ΣΙΙα) (Xiao et al., 2016), the degree of cyclisation of tetramethylated brGDGTs (#ringstetra) (Sinninghe Damsté, 2016) and comparison of the relative abundance of tetramethylated, pentamethylated and hexamethylated brGDGTs (Sinninghe Damsté, 2016).

The ΣΙΙΙα/ΣΙΙα ratio was derived from a global dataset comprising 1,354 terrestrial and 589 marine samples. Notably, 90% of marine sediments exhibited a ΣΙΙΙα/ΣΙΙα >0.92, while 90% of terrestrial sediments had ΣΙΙΙα/ΣΙΙα of <0.59 (Xiao et al., 2016). In their study, Xiao et al. (2016) combined IIIa and IIIa' and IIa and IIa' as the majority of the available data at that time did not distinguish between the isomers. As a result, the proposed proxy was predominantly based on data lacking isomer separation. With improved compound separation achieved by the HPLC method of Hopmans et al. (2016), now both 5- and 6- methyl brGDGTs are incorporated in the calculation.

The #rings<sub>tetra</sub> approach is based on the comparison between the global soil dataset and sediments from a variety of open sea, coastal and river environments, which shows that the #rings<sub>tetra</sub> value in soils is always <0.7, suggesting that brGDGTs in marine sediments with values >0.7 have at least in part a marine origin (Sinninghe Damsté, 2016). It was also observed that the relative abundance of tetra-, penta- and hexamethylated brGDGTs in soils followed a clear trend when plotted in a triplot, and that datapoints derived from (coastal) marine sediments plot increasingly offset from this trend depending on the contribution of *in situ* produced brGDGTs in marine sediments (Sinninghe Damsté, 2016). The observed discrepancies in the degree of methylation and cyclisation of brGDGTs in marine sediments and soils have been attributed to pH differences between soils and marine waters, and to lower temperatures in the deep ocean than in soils (Sinninghe Damsté, 2016; Xiao et al., 2016). These new approaches supplement the use of the BIT index as a tracer of terrestrial brGDGT inputs to marine sediments (see Section 6.1), particularly in coastal regions where primary productivity is controlled mainly by nutrient input from rivers, such as, for example, in Chinese coastal seas (Liu et al., 2021). In this configuration, the increase in marine isoGDGT production due to increased *Nitrososphaera* productivity offsets the terrestrial brGDGTs inputs from the rivers, resulting in a low BIT index despite high inputs from land (Liu et al., 2021).

Identifying the source of brGDGTs in marine environments is crucial for applying both terrestrial or marine paleothermometers. In the case of significant *in situ* marine production of brGDGTs in marine settings, brGDGT-based temperature proxies for air temperatures (such as the MBT'<sub>5ME</sub> index) should be applied with care (cf. De Jonge et al., 2014b; see also Inglis et al., 2023), or corrected for the marine contribution prior to reconstructing mean annual air temperatures from marine sediments (Dearing-Crampton-Flood et al., 2018). Although *in situ* production of brGDGTs in marine environments complicates their use as proxies for terrestrial environmental conditions, recent studies suggest that their distribution can provide information about other marine environmental conditions, such as oxygen conditions (Liu et al., 2014; Xiao et al., 2024).

### 8.4 GTGTs, GMGTs and GDDs







Glycerol trialkyl glycerol tetraethers (GTGTs) and glycerol dialkanol diethers (GDDs) have been identified in cultured archaea (e.g., Elling et al., 2014, 2017; Bauersachs et al., 2015) and marine sediments (Liu et al., 2012b, 2018; Xu et al., 2020). GDD have been speculated to be either biosynthetic intermediates (e.g., Meador et al., 2014) or degradation products (Coffinet et al., 2015). In fact, a degradation pathway was proposed by Liu et al. (2016) suggesting that isoGDDs are formed from isoGDGTs. Recent discoveries in the isoGDGT biosynthetic pathway (Zeng et al., 2019; Lloyd et al., 2022) do not consider the involvement of GDDs as intermediates in the biosynthesis, giving further support to GDDs as degradation products. Alternatively, it has been shown that GDDs could be degradation products of GDGTs (Coffinet et al., 2015; Mitrović et al., 2023; Hingley et al., 2024). Glycerol monoalkyl glycerol tetraethers (GMGTs) were first identified in a hyperthermophilic methanogen (Morii et al., 1998), but later appeared to also occur in sediments from low-temperature marine and lacustrine environments (e.g., Schouten et al., 2008; Liu et al., 2012a), where they were inferred to possibly be derived from Euryarchaeota. Their relative increase with temperature in marine hydrothermal sediments suggests that they may play a role in thermal regulation for their archaeal source organism (Sollich et al., 2017; Hernández-Sánchez et al., 2024), as recently supported by a mechanism linking GMGTs to high temperatures using molecular dynamics simulations (Garcia et al., 2024; Zhou and Dong, 2024). Next to isoGMGTs, branched GMGTs (brGMGTs) also exist, and are found in marine sediments of modern (Liu et al., 2012a) to late Cretaceous (e.g., Bijl et al., 2021), where their distributions, including methylation, strongly, but not consistently, vary in response to environmental change, likely temperature and/or water column oxygenation (Sluijs et al., 2020; Bijl et al., 2021; Kirkels et al., 2022) although in all these applications, brGMGTs have a different relationship to temperature. BrGMGTs furthermore occur in oxygen minimum zone SPM from the eastern Pacific (Xie et al., 2014), which agrees with the identification of the enzyme that synthesizes GMGTs, which is associated with obligate anaerobic archaea in oxygen-deficient (O<sub>2</sub> < 25 μM) environments (Li et al., 2024). While GTGTs, GDDs, and GMGTs are currently not commonly investigated or reported in marine sediments, future research may explore their potential applications in paleoclimate studies.

## 9 Best practices for sample, site and proxy intercomparisons and the presentation of error and uncertainty

Studies using GDGTs for temperature reconstruction can span or compile multiple sites (e.g., O'Brien et al., 2020; Auderset et al., 2022; Hou et al., 2023), or different depositional settings within an individual study site, e.g., fully marine to shallow marine or glaciomarine (e.g., Śliwińska et al., 2019; Duncan et al., 2022). It is important to consider variability in the depositional setting through a record, or in multi-site compilations, as this can influence GDGT preservation, or the water depth GDGTs have been exported from (e.g., Huguet et al., 2008; Taylor et al., 2013; Duncan et al., 2022). Integration of GDGT distributions and screening indices with the wider sedimentological and depositional history of a site should play an

important role in interpreting a GDGT record. Likewise, other proxy methods for environmental reconstruction, such as those based on microfossils or other molecular fossils, can serve to support or help interpret a GDGT temperature record.







While the analytical error in GDGT-based temperature proxies is small (see Figure 2), the calibration error imposes uncertainty on the absolute reconstructed SST values. This calibration error includes variability caused by known factors which could bias TEX<sub>86</sub> and its relationship to temperature, such as overprints, water column factors etc. One way to reduce the calibration error is by more careful selection of subsets from the surface sediment calibration that, based on what we now know, most accurately reflects temperature (e.g., as in Fokkema et al., 2024). In any case, we recommend that graphs wherein GDGT-based temperature records are presented should indicate the calibration error. We note that the calibration error is occasionally taken to represent the uncertainty in the reconstructed SST from sample to sample, and interpreted to mean that only SST shifts larger than the calibration error can be interpreted from any proxy record, whereas this error addresses the uncertainty of the record as a whole. In fact, many available reconstructions show predictable (e.g., relative to other temperature proxies) sampleto-sample TEX<sub>86</sub> variability, well within the range of the calibration error (e.g., Bijl et al., 2021; Hou et al., 2023). This distinction is important, because despite some level of uncertainty in the absolute temperature reconstruction derived from GDGTs, the temperature trends and sample-to-sample variability within the calibration error still hold paleoceanographic significance. One way to assess the significance of predictable or expected (e.g., relative to other temperature proxies or sites subject to similar conditions) temperature trends is to examine downcore variations in temperature proxy values and compare them with the analytical error before proxy conversion into SST. Indeed, the cumulative effect of all non-thermal effects at the spatial scale (e.g., global or regional) of the selected calibration may not apply to one specific site, so the calibration error should be viewed as an upper-bound of the uncertainty attributable to the non-thermal effects relevant to this site (Daytian et al., 2019). Therefore, applying the calibration error as uncertainty on all samples (e.g., via an envelope, or by a temperature error bar on each sample) gives the false impression that downcore trends within the calibration error might not be significant. We therefore recommend visualizing the calibration error separate from the individual data points, e.g., as bars in the corner of the plot.

### 10 Data reporting and archiving

## 10.1 Towards a common approach in GDGT data reporting and archiving

Since the development of the UHPLC-MS technique for analysis of GDGTs (Hopmans et al., 2000), research into the environmental occurrence of GDGTs synthesized by archaea and bacteria has greatly expanded. To date, tens of thousands of GDGT abundances determined from laboratory experiments (cultures and mesocosms), modern environmental archives (marine waters and sediments), and ancient sedimentary sequences, have been reported in published literature. Additionally,

the recent improvement of analytical techniques and methodology in lipid determination allowed scientists to discover newer classes of GDGT compounds.




Published temperature records derived from GDGTs are largely accompanied by raw data. Researchers commonly use online archiving systems like World Data Service for Paleoclimatology and PANGAEA to make their data publicly available. However, data reporting and accessibility have never been fully systematized. To date, there is no agreed-upon standard for reporting GDGT information upon publication. As the number of GDGT measurements carried out by the community has grown over the years, the effort to analyze, integrate, and/or synthesize such large datasets requires a significant amount of time to manually compile and vet each data record individually (cf. Judd et al., 2022; PhanSST database). In this section, we aim to provide a list of recommended items for reporting GDGT information in publications. The recommendations are based on the Linked Paleo Data (LiPD; Mckay and Emile-Geay, 2016) data standards and architecture, ensuring adherence to FAIR open access principles and facilitating the interchange.

# 1145 10.2 Data components

Following LiPD data reporting framework, six possible types of information should be included when reporting GDGT data in any publications, including (1) root metadata, (2) geographic metadata, (3) publication metadata, (4) funding metadata, (5) paleodata information, and (6) geochronological information (e.g., age-depth models). Scientific journals and/or online archiving systems may require some data components during the peer review process, making sure to have all the data components listed here available at publication will ensure the integrity of the data for future use. Table 4 provides further descriptions of each data component.

Table 4: Description of data components needed to be considered when reporting GDGT information. Please note that the list provided here is not exhaustive.

| No. | Data<br>Component                                                                                                                                                                                                                                                                                                                                                                                                                                                                                                                                                                                                                                                                                                                                                                                                                                                                                                                                                                                                                                                                                                                                                                                                                                                                                                                                                                                                                                                                                                                                                                                                                                                                                                                                                                                                                                                                                                                                                                                                                                                                                                              | Description(s)                                                                                                                                                                                                     | Eleme | nts in each component                                        | Level of Importance<br>(Required, Preferred, Optional)                                                                                                  |
|-----|--------------------------------------------------------------------------------------------------------------------------------------------------------------------------------------------------------------------------------------------------------------------------------------------------------------------------------------------------------------------------------------------------------------------------------------------------------------------------------------------------------------------------------------------------------------------------------------------------------------------------------------------------------------------------------------------------------------------------------------------------------------------------------------------------------------------------------------------------------------------------------------------------------------------------------------------------------------------------------------------------------------------------------------------------------------------------------------------------------------------------------------------------------------------------------------------------------------------------------------------------------------------------------------------------------------------------------------------------------------------------------------------------------------------------------------------------------------------------------------------------------------------------------------------------------------------------------------------------------------------------------------------------------------------------------------------------------------------------------------------------------------------------------------------------------------------------------------------------------------------------------------------------------------------------------------------------------------------------------------------------------------------------------------------------------------------------------------------------------------------------------|--------------------------------------------------------------------------------------------------------------------------------------------------------------------------------------------------------------------|-------|--------------------------------------------------------------|---------------------------------------------------------------------------------------------------------------------------------------------------------|
| 1   | Root<br>Metadata<br>(Dataset<br>Metadata)                                                                                                                                                                                                                                                                                                                                                                                                                                                                                                                                                                                                                                                                                                                                                                                                                                                                                                                                                                                                                                                                                                                                                                                                                                                                                                                                                                                                                                                                                                                                                                                                                                                                                                                                                                                                                                                                                                                                                                                                                                                                                      | This contains basic information of the reporting dataset, including but not limited to:                                                                                                                            | i     | Dataset name                                                 | Required                                                                                                                                                |
|     |                                                                                                                                                                                                                                                                                                                                                                                                                                                                                                                                                                                                                                                                                                                                                                                                                                                                                                                                                                                                                                                                                                                                                                                                                                                                                                                                                                                                                                                                                                                                                                                                                                                                                                                                                                                                                                                                                                                                                                                                                                                                                                                                |                                                                                                                                                                                                                    | ii    | Author(s)/Investigator(s)                                    | Required                                                                                                                                                |
|     |                                                                                                                                                                                                                                                                                                                                                                                                                                                                                                                                                                                                                                                                                                                                                                                                                                                                                                                                                                                                                                                                                                                                                                                                                                                                                                                                                                                                                                                                                                                                                                                                                                                                                                                                                                                                                                                                                                                                                                                                                                                                                                                                |                                                                                                                                                                                                                    | iii   | Sample request number(s)/ID(s)                               | Required                                                                                                                                                |
|     |                                                                                                                                                                                                                                                                                                                                                                                                                                                                                                                                                                                                                                                                                                                                                                                                                                                                                                                                                                                                                                                                                                                                                                                                                                                                                                                                                                                                                                                                                                                                                                                                                                                                                                                                                                                                                                                                                                                                                                                                                                                                                                                                |                                                                                                                                                                                                                    | iv    | Cruise name/ID                                               | Required                                                                                                                                                |
|     |                                                                                                                                                                                                                                                                                                                                                                                                                                                                                                                                                                                                                                                                                                                                                                                                                                                                                                                                                                                                                                                                                                                                                                                                                                                                                                                                                                                                                                                                                                                                                                                                                                                                                                                                                                                                                                                                                                                                                                                                                                                                                                                                |                                                                                                                                                                                                                    | V     | Link(s) to published dataset(s)                              | Required - This information will be available after the author(s) publishing the dataset with an online archive, such as <u>PANGAEA online database</u> |
| 2   | Geographic<br>Metadata                                                                                                                                                                                                                                                                                                                                                                                                                                                                                                                                                                                                                                                                                                                                                                                                                                                                                                                                                                                                                                                                                                                                                                                                                                                                                                                                                                                                                                                                                                                                                                                                                                                                                                                                                                                                                                                                                                                                                                                                                                                                                                         | This contains geographical information of study sites, including but not limited to:                                                                                                                               | i     | Coordinate(s) (modern latitude/longitude)                    | Required                                                                                                                                                |
|     |                                                                                                                                                                                                                                                                                                                                                                                                                                                                                                                                                                                                                                                                                                                                                                                                                                                                                                                                                                                                                                                                                                                                                                                                                                                                                                                                                                                                                                                                                                                                                                                                                                                                                                                                                                                                                                                                                                                                                                                                                                                                                                                                |                                                                                                                                                                                                                    | ii    | Site name(s)                                                 | Required                                                                                                                                                |
|     |                                                                                                                                                                                                                                                                                                                                                                                                                                                                                                                                                                                                                                                                                                                                                                                                                                                                                                                                                                                                                                                                                                                                                                                                                                                                                                                                                                                                                                                                                                                                                                                                                                                                                                                                                                                                                                                                                                                                                                                                                                                                                                                                |                                                                                                                                                                                                                    | iii   | Descriptive information such as:  • Country, State, Province | Preferred - especially for studies with samples from geological outcrops                                                                                |
|     |                                                                                                                                                                                                                                                                                                                                                                                                                                                                                                                                                                                                                                                                                                                                                                                                                                                                                                                                                                                                                                                                                                                                                                                                                                                                                                                                                                                                                                                                                                                                                                                                                                                                                                                                                                                                                                                                                                                                                                                                                                                                                                                                |                                                                                                                                                                                                                    |       | Ocean basin/region                                           | Optional - for marine samples, providing site names with coordinates is sufficient                                                                      |
| 3   | Metadata information informati | This contains publication information of GDGT data retrieved from previously published datasets, including but not limited to:                                                                                     | i     | Author(s)                                                    | Required                                                                                                                                                |
|     |                                                                                                                                                                                                                                                                                                                                                                                                                                                                                                                                                                                                                                                                                                                                                                                                                                                                                                                                                                                                                                                                                                                                                                                                                                                                                                                                                                                                                                                                                                                                                                                                                                                                                                                                                                                                                                                                                                                                                                                                                                                                                                                                |                                                                                                                                                                                                                    | ii    | Title                                                        | Preferred                                                                                                                                               |
|     |                                                                                                                                                                                                                                                                                                                                                                                                                                                                                                                                                                                                                                                                                                                                                                                                                                                                                                                                                                                                                                                                                                                                                                                                                                                                                                                                                                                                                                                                                                                                                                                                                                                                                                                                                                                                                                                                                                                                                                                                                                                                                                                                |                                                                                                                                                                                                                    | iii   | Journal name                                                 | Optional                                                                                                                                                |
|     |                                                                                                                                                                                                                                                                                                                                                                                                                                                                                                                                                                                                                                                                                                                                                                                                                                                                                                                                                                                                                                                                                                                                                                                                                                                                                                                                                                                                                                                                                                                                                                                                                                                                                                                                                                                                                                                                                                                                                                                                                                                                                                                                | Not all information will be required, but the authors need to make sure that the publication metadata is sufficient for readers to be able to track back to the original publications of the compiled information. | iv    | DOI                                                          | Preferred                                                                                                                                               |
|     |                                                                                                                                                                                                                                                                                                                                                                                                                                                                                                                                                                                                                                                                                                                                                                                                                                                                                                                                                                                                                                                                                                                                                                                                                                                                                                                                                                                                                                                                                                                                                                                                                                                                                                                                                                                                                                                                                                                                                                                                                                                                                                                                |                                                                                                                                                                                                                    | V     | Year of publication                                          | Required                                                                                                                                                |
|     |                                                                                                                                                                                                                                                                                                                                                                                                                                                                                                                                                                                                                                                                                                                                                                                                                                                                                                                                                                                                                                                                                                                                                                                                                                                                                                                                                                                                                                                                                                                                                                                                                                                                                                                                                                                                                                                                                                                                                                                                                                                                                                                                |                                                                                                                                                                                                                    | vi    | Link(s) to original publication(s)                           | Optional                                                                                                                                                |

| No. | Data<br>Component                                                                                        | Description(s)                                                                   | Elements in each component |                                                                                                                                                                                                             | Level of Importance<br>(Required, Preferred, Optional)                                                                                                              |
|-----|----------------------------------------------------------------------------------------------------------|----------------------------------------------------------------------------------|----------------------------|-------------------------------------------------------------------------------------------------------------------------------------------------------------------------------------------------------------|---------------------------------------------------------------------------------------------------------------------------------------------------------------------|
| 4   | 4 Funding This is applicable when the research that produced the data was funded. The metadata includes: | i                                                                                | Funding agency             | Generally required by journal(s)/publisher(s)                                                                                                                                                               |                                                                                                                                                                     |
|     |                                                                                                          |                                                                                  | ii                         | Funding grant number(s)/ID(s)                                                                                                                                                                               |                                                                                                                                                                     |
| 5   | Information paleoenvironm including but r  Note: GDGT a material inform commonly rep                     | All measured and inferred paleoenvironmental data, including but not limited to: | GDG1                       | Abundances                                                                                                                                                                                                  |                                                                                                                                                                     |
|     |                                                                                                          |                                                                                  | i                          | Raw peak intensities including the standard                                                                                                                                                                 | Required                                                                                                                                                            |
|     |                                                                                                          | Note: GDGT abundances and material information are commonly reported in data     | ii                         | Absolute abundances (required material information)                                                                                                                                                         | Optional                                                                                                                                                            |
|     |                                                                                                          | tables and/or spreadsheets.                                                      | iii                        | Fractional abundances                                                                                                                                                                                       | Optional                                                                                                                                                            |
|     |                                                                                                          |                                                                                  |                            |                                                                                                                                                                                                             | If reported, the authors MUST explicitly describe all the fractions used for the fractional abundance calculation, i.e., all fractions that will give the sum to 1. |
|     |                                                                                                          |                                                                                  | Mater                      | ial Information                                                                                                                                                                                             |                                                                                                                                                                     |
|     |                                                                                                          |                                                                                  | iv                         | Sample information, including:  For IODP samples  Site, Hole, Core, section, interval, depth (and which depth scale), age (and whose age model)  For non-IODP/outcrop samples  Hole/Core/S ection/Interv al | Required                                                                                                                                                            |
|     |                                                                                                          |                                                                                  | V                          | Sample weight(s)                                                                                                                                                                                            | Required when reported absolute abundances                                                                                                                          |
|     |                                                                                                          |                                                                                  | vi                         | Amount of spiked standard                                                                                                                                                                                   | Required when reported absolute abundances                                                                                                                          |

| No. | Data<br>Component                   | Description(s)                                                              | Elements in each component |                                                                                                         | Level of Importance<br>(Required, Preferred, Optional)                                                                                  |
|-----|-------------------------------------|-----------------------------------------------------------------------------|----------------------------|---------------------------------------------------------------------------------------------------------|-----------------------------------------------------------------------------------------------------------------------------------------|
|     |                                     |                                                                             | vii                        | Sample description(s) (usually optional): Colour, texture, key features from core images                | Optional                                                                                                                                |
|     |                                     |                                                                             | Samp                       | le Preparation Information                                                                              |                                                                                                                                         |
|     |                                     |                                                                             | vii                        | Sample preparation information<br>Lipid extraction and purification<br>method(s) used                   | Required - usually described in the "Methods and Materials" section in scientific reports/publications. See Section 10.2.2 for details. |
| 6   | Geochronolo<br>gical<br>Information | This contains information used to infer the age of individual GDGT samples. | i                          | Age-depth models  Tie points  Age determination approach  Linear interpolation between tie points  etc. | Required if sample age is reported.                                                                                                     |

#### 10.2.1 Non-GDGT information (metadata)






This section includes a list of recommended metadata of non-GDGT information to be reported alongside GDGT data. We have categorized the importance of each metadata item into three levels: required, recommended, and optional. Required information must be publicly accessible at time of publication. Additionally, we discuss best practices for reporting this information and provide guidance on handling missing information from published literature when necessary.

- Sample Name and/or Sample ID (required): Samples must be identified with unique and unambiguous sample names and/or sample identifications (IDs). Crucially, this label must provide the explicit link between published data and vial/sampleID. For example, the double-column development in GDGT analyses on UHPLC-MS (Hopmans et al., 2016) required that previously measured GDGTs had to be re-measured, highlighting the importance of proper sample labeling and storage. GDGTs can be stored for decades in their vials, which yields an opportunity for future remeasurements when labeling is adequate and links to original datasets.
- Sample Request Number and/or Sample Request ID (required when applicable, in e.g., IODP regime): This provides a direct way to link GDGT data with the original source of the sample information that is curated at the core repository.
- Sample Information (required). The majority of GDGT research is done on core material, for instance from the IODP regime and its predecessors, but also from piston coring