# Peer review of "Reviews and Syntheses: Best practices for the application of marine GDGTs as proxy for paleotemperatures: sampling, processing, analyses, interpretation, and archiving protocols"

_EGUsphere, 2025_

## Author Comment (AC1)

Bijl and co-authors present a comprehensive review of GDGTs, with focus on marine isoprenoid GDGTs, and their consensus on best practices for sampling, processing, analyses, interpretation, and archiving. This is an impressive effort that transcends the typical review focused on research-focused questions. I greatly appreciated the detailed compilation of sample prep, processing and analytical considerations, which will be useful to anyone new to the field or simply in need of a handy reference in their lab. The archiving section is also timely given the more recent shift to open science and big data synthesis efforts. Following some minor revisions and considerations, I think this paper will be a valuable addition to the GDGT (and biomarker) literature.

--David Harning

**Reply: We thank the reviewer for the positive assessment of our manuscript. Below we indicate with each comment how we intend to adjust the manuscript.**

General: Structure could be improved in places. Sometimes paragraphs are really short and others overly long. To improve flow in a monster paper like this, it is important to try to stick with topic and concluding sentences for paragraphs to enhance the flow and readability for the reader.

**Reply: We thank the reviewer for this comment. We will critically review the text of the manuscript and make sure the paragraphs are equal in flow and pace.**

Title: While GDGTs are most often used to reconstruction temperature, as authors note in the text, other environmental factors are known to influence their distributions. Recently, it has been exciting to see efforts to use GDGT distributions to reconstruct these other processes (e.g., archaeal ecology, nutrients, etc.). Hence, I may suggest that "paleotemperatures" is changed to "paleoenvironments". Or somehow recognize these other factors. Otherwise, I feel the authors are pigeonholing this proxy into just temperature, which we know is rarely the only factor.

**Reply: We thank the reviewer for highlighting the other uses of GDGTs as proxies for other environmental parameters other than temperature. The reviewer is correct to note the other uses of GDGTs than for temperature, and this is stressed in the manuscript already under section 6 which describes the possible non-thermal factors and overprints. However, we acknowledge that a) the most dominant factor in GDGT distributions is temperature, highlighted by numerous studies/publications, b) the application of GDGTs as proxy by the (paleo)climate community is predominantly focusing for paleotemperature reconstruction, and c) it was the explicit ambition of this paper to focus on that application. We feel that the title as it was gives proper credit to our focus. However, we will highlight more explicitly the application of GDGTs as proxy for other environmental parameters in the introduction, to alleviate the reviewers' concerns.**

Specific points:

L103-104: Change to's to with's

**Reply: We will change this accordingly**

L111: I suggest a new paragraph here on brGDGTs. Bump the first sentence on isoGDGTs to the next paragraph where those compounds are being discussed.

**Reply: We will change this accordingly**

L121-130: This paragraph on GDGTs could use some reference to the other factors that may influence their distribution besides temperature, as discussed in detail later in the text. Having this upfront will hopefully ensure GDGT users remain cautious with temperature interpretations.

**Reply: We will change this accordingly**

L131: The final sentence on OH-GDGTs could become its own new paragraph with a similar level of discussion on environmental considerations as given to the branched and isoprenoid GDGTs before.

**Reply: We will adjust this paragraph in line with the reviewers' suggestions**

L139: I would suggest given the uncertainties around what temperature or water depth GDGTs may capture, you leave it as past ocean temperature rather than sea surface temperature

**Reply: We will change this accordingly**

L142: remove and

**No change needed, as no redundant "and" was observed in this sentence**

Figure 1: Only because they are my own, I noticed that Baffin Bay core top samples are not included in the map (Harning et al., 2023). Please make sure that any other sites that may be missing are included as well.

**Reply: We will double check for missing data and update the figure**

L162: Instead of "right kind", perhaps change to needed. Right kind sort of implies there is a wrong kind or bad type of data, which I don't think is the intent

**Reply: We deleted "right kind"**

L163: The PhanSST example is given but another sentence could be added to better demonstrate how this example serves the authors point

**Reply: We will add a sentence summarizing how this effort serves the point we intend to make**

L164: remove greatly – it's too relative of a term and doesn't mean much

**Reply: We will change this accordingly**

L165-166: I agree this is important and as I note at the end, perhaps foreshadowing some next steps would be helpful. May not be needed here in the introduction though. I suggest maybe in lieu of conclusions at the end – see final comment

**Reply: see reply to the final comment**

L168-176: This list appears abruptly and seems out of place. If you wish to keep, it needs some intro text. I think it could be removed though.

**Reply: we agree that this list is out of context of the text**

**Proposed changes: we will restructure the text so that this list becomes a clear introduction to the structure of the paper**

L187: change this to these

**Reply: We will change this accordingly**

L188: subsurface "sediment"? Also please clarify what shallow means if there are any numbers

**Reply: We will clarify in the text**

L201: remove really

**Reply: We will change this accordingly**

L208: How do we know when the outcrop sediment is not affected by weather? How deep do we need to excavate to ensure samples are optimal?

**Reply: We will add some indications to this, based on the literature**

L226: Instead of sea above, simply reiterate the temperature threshold again for the reader

**Reply: We will change this accordingly**

L233: Regarding coring contamination, it's also important to note that we do this as the core tubing itself is often some form of plastic that should be avoided for potential contamination too. Also important to consider because the internal sediments are least deformed by coring

**Reply: This point was already addressed 3 lines further down in the original manuscript**

L239: I think you're missing the age here; the outcrop is modern right?

**Reply: We will make the age more explicit**

L251: SIM is not always used

**Reply: We will add this nuance**

L258-260: Most of this seems better placed with the sample processing overview. Maybe just keep ones relevant for field and sub-sampling? Same for following paragraph

**Reply: Good point, we placed these two paragraphs in Section 3.7**

L303: Our lab does 2x, I see 3x a lot, but based on our analyses, the majority (sorry I don't have numbers in front of me) is extracted with 2. Including either of both numbers seem helpful here. Also, might be worth noting that cleaning the ASE with pure methanol (or some other polar solvent) rinses between batches is helpful to avoid cross-contamination

**Reply: We will add this nuance**

L356: IPL extractions could be its own section, so it's not lost in here with ultrasonic

**Reply: We will follow this advice**

L369: This paragraph seems applicable to all extraction methods so maybe not include under ultrasonic. May also be worth noting to not dry down to completion under N2 gas to avoid losing more volatile compounds

**Reply: We will follow this advice. We moved up the final paragraph to section 3.3.**

L394: I think this paragraph should be joined with the previous

**Reply: We will change this accordingly**

L412: remove there

**Reply: We will change this accordingly**

L487: not always, we use full scan with an orbitrap and then extract ion chromatograms

**Reply: We will add this nuance**

Figure 2: This is really cool data to see!

**Reply: Thank you!**

L673: This isoGDGT-0/cren value of 2, based on my understanding, is completely arbitrary and I'm not convinced we have a good threshold for this ratio and how it relates to dominant GDGT producers. Culture experiments would help. Would be good to articulate if similar for other thresholds discussed in this review.

**Reply: We agree that these thresholds are probably somewhat setting-specific. We will address this nuance here and at the other indices**

Table 2: Some equations are cut off. Also please provide references for status column recommendations if not already included.

**Reply: We will fix this formatting issue**

L908: This is an example of a paragraph that lacks appropriate structure with e.g., topic sentences. It is rather long and could be revised to improve clarity.

**Reply: We will restructure this paragraph so that it is in line with the flow of the rest of the paper**

L927: What is the "GDGT review paper"? Reference?

**Reply: this is a paper that is currently in preparation. We will remove this reference since that paper is not yet findable**

Table 3: Some equations are cut off

**Reply: We will fix this formatting issue**

L1089: Remove very - it's too relative of a term and doesn't mean much

**Reply: this will be removed**

L1120: World Data "Service" if you are referring to NOAA

**Reply: We will change this accordingly**

L1243: The conclusions are quite brief, which is maybe okay given the details included in the manuscript and that this is a review. That being said, I miss some sort of synopsis on what the possible next steps may be. This could be included here instead of conclusions. How do we ensure community buy-in and the incorporation of the entire GDGT community? I understand this review came out of the GDGT workshop, but many folks, including myself, could not attend due to cost, distance, etc. How do we make this more inclusive for people from outside Europe, who were the main attendees for the workshop, and from other often less represented regions of the globe (e.g., global south)? In this sense, I would not say this represents a "large" part of the lipid biomarker community. For next steps, virtual workshops seem like a good opportunity and perhaps some sort of GDGT-specific database that incorporates the metadata fields highlighted in the final section. Just some things to think about that might help this review helpful in implementing the key take-aways from this paper.

**Reply: we thank the reviewer for this thoughtful comment. Our revised manuscript will have a section added with steps forward to work on even better workflow, reproducibility and optimization of data. We cannot speculate in the rebuttal or in the paper on the scope and logistical setup of potential future**

**workshops, as it would put too much pressure on something that has not developed yet. Naturally the ambition must be that attendance and participation of any future workshop should be as inclusive and open as possible.**

---

## Author Comment (AC2)

The work by Bijl et al. presents an overview of the state of the art of the analysis and interpretation of GDGTs in the marine environment. In this review the authors discuss the processing of GDGT related information from the sample collection, analysis, to ultimately the long-term storage of the information for future use, following the FAIR principles. Along the text the authors discuss both the history, current state, as well as considerations for each of the steps. I find the review very engaging and easy to read, with the structure of the text accessible for people that might want to consult specific details on the topic.

I am happy to recommend this manuscript for publication once some minor comments are addressed.

**Reply: we thank the reviewer for the positive assessment, and for the constructive comments that will improve the paper. Below we respond to each comment, how we intend to improve the manuscript.**

Line 107. Change "Next to these". Additionally, the list of compounds in this paragraph breaks the flow of the idea. Consider breaking the paragraph.

**Reply: we will change this accordingly**

Line 110 (and others). My recommendation would be to stick with "GMGTs" as that's the most descriptive name. For clarity I understand making the clarification, but later they should be referred as GMGTs.

**Reply: we will change this accordingly**

Lines 139. I am not sure if IPLs are mentioned enough in the rest of the text to include them in this sentence.

**Reply: We will remove IPLs from this sentence**

Line 258. Correct this section: "all metal tools.extraction"

**Reply: we will make the section correct**

Section 3.2. This section would benefit from some references.

**Reply: we will add relevant references**

Line 329. I don't know if I would make emphasis on Soxhlet being used for "larger samples" when the described range is very close to that of ASE.

**Reply: agreed. The advantage of Soxhlet over other techniques is the cleanliness and completeness of the extraction. We will highlight this information in this section.**

Line 345. Consider starting a new section or starting a new paragraph here.

**Reply: we will adjust this so that the paragraph matches the flow of the rest of the paper better**

Line 348. Here the Bligh-Dyer method is mentioned but only a very brief description is given, and it is not until a paragraph later that some information about this protocol is given. Maybe it would be worth explaining it within the Ultrasonic extraction section.

**Reply: we move the Bligh-Dyer protocol within the section of Ultrasonic extraction, and remove repetition of that in the paragraphs below**

Line 373. Make the 2 underscore in N2.

**Reply: we will correct this formatting error**

Lines 547-548. I am not sure why the temperatures are given in F here.

**Reply: we thank the reviewer for spotting this. We will remove Fahrenheit and keep Celsius**

Line 576. I think this should be Figure 3f?

**Reply: yes, we will correct this error**

Section 6.7. Maybe the explanation of how OPTiMAL could be improved, since it is a bit hard to follow.

**Reply: we will make sure that the description of OPTiMAL is consistent with the flow of the rest of the paper, and improve the readability.**